



# Natural and anthropogenic influence on tropospheric ozone variability over the Tropical Atlantic unveiled by satellite and in situ observations

Sachiko Okamoto[1, a], Juan Cuesta[1], Gaëlle Dufour[2], Maxim Eremenko[1], Kazuyuki Miyazaki[3], Cathy Boonne[4], Hiroshi Tanimoto[5], Jeff Peischl[6] and Chelsea Thompson[6]

[1] Univ Paris Est Créteil and Université de Paris Cité, CNRS, LISA, F-94010 Créteil, France

[2] Université de Paris Cité and Univ Paris Est Créteil, CNRS, LISA, F-75013 Paris, France

[3] Jet Propulsion Laboratory (JPL), California Institute of Technology, Pasadena, 91109 CA, USA

[4] Institut Pierre Simon Laplace (IPSL), AERIS data centre, Paris, 75252, France

[5] National Institute for Environmental Studies, Tsukuba, 350-8506, Japan

[6] NOAA Chemical Sciences Laboratory, Boulder, CO, 80305, USA

[a] now at : LATMOS, Sorbonne Université, UVSQ, CNRS, Paris, 75252, France

*Correspondence to*: Sachiko Okamoto (sachiko.okamoto@latmos.ipsl.fr) and Juan Cuesta (cuesta@lisa.ipsl.fr)

**Abstract.** Tropospheric ozone over the South and Tropical Atlantic plays an important role in the photochemistry and energy budget of the atmosphere. In this remote region, tropospheric ozone estimates from reanalysis datasets show the largest discrepancies. The present study characterises the vertical and horizontal distribution of tropospheric ozone over the South and Tropical Atlantic during February 2017 using a multispectral satellite approach called IASI+GOME2 and in situ airborne measurements from the Atmospheric Tomography Mission (ATom). These observations are compared with three global chemistry reanalysis products: the Copernicus Atmosphere Monitoring Service reanalysis (CAMS reanalysis), the Tropospheric Chemistry Reanalysis version 2 (TCR-2), and the second Modern-Era Retrospective Analysis for Research and Applications (MERRA-2). The CO-enriched air masses from Western and Central Africa are lifted into the middle and upper troposphere over the ocean by strong upward motions. In the descending branches of the Hadley cells over the Southern Atlantic, stratospheric intrusions are observed. Air masses in the Southern Hemisphere are influenced by biomass burning sources from Central and Eastern Africa and lightning, as well as downdrafts from the stratosphere. According to in situ measurements of chemical tracers, tropospheric ozone attributed to biomass burning emissions of ozone precursors is approximately 13 ppb (~17 %) over 7 km (25° S–5° N) and approximately 38 ppb (~50 %) over 3 km (25° S–15° S). The intercomparison suggests a significant overestimation of three chemistry reanalysis products of lowermost troposphere ozone





over the Atlantic in the Northern Hemisphere because of the overestimations of ozone precursors from anthropogenic sources
from North America.

## 1 Introduction

Tropospheric ozone is one of the key gases in the atmosphere because it is a major greenhouse gas (Szopa et al., 2021) and it plays an important role in determining the oxidising capacity of the troposphere (Monks et al., 2015). The main source of tropospheric ozone is in situ photochemical production via oxidation of non-methane volatile organic compounds (NMVOCs), carbon monoxide (CO), and methane ($CH_4$), in the presence of nitrogen oxides ($NO_X$) (e.g., Atkinson, 2000). Stratosphere-to-troposphere transport of ozone also significantly increases its abundance in the troposphere (e.g., Stohl et al., 2003). The lifetime of ozone in the troposphere ranges from a few hours in polluted urban areas to up to a few weeks in the free troposphere, but is relatively long on average (~22 days; Young et al., 2013). This allows tropospheric ozone to be transported over distances of intercontinental and hemispheric scales.

The South and Tropical Atlantic is one of the regions with the sparsest coverage of in-situ observations, while being influenced by large amounts of ozone precursors from biomass burning (van der Werf et al., 2017), biogenic (Sindelarova et al., 2022) and lightning (Schumann and Huntrieser, 2007) over the African and the South American continents. In-Service Aircraft for a Global Observing System (IAGOS) is a European Research Infrastructure using commercial aircraft for measurements of atmospheric compositions including ozone and CO (e.g. Petzold et al., 2015). This programme builds on the heritage of the former research projects the Measurement of Ozone and Water Vapour on Airbus In-service Aircraft (MAZAIC) and the Civil Aircraft for the Regular Investigation of the Atmosphere Based on an Instrument Container (CARIBIC), proving valuable datasets over the Atlantic Ocean. Previous studies have shown the tropical ozone distributions by using these datasets (Lannuque et al., 2021; Sauvage et al. 2005; 2007b; Tsivlidou et al., 2023; Yamasoe et al., 2015). Their results show for example that lightning has the largest influence on the South Atlantic ozone burden (24° S–0°, 35° W–10° E), accounting for more than 37 % (Sauvage et al. 2007a). The authors quantified that the contributions of biomass burning, fossil fuel combustion, and soil $NO_X$ emissions to the tropospheric ozone column were 6, 5, and 4 times smaller than that from lightning. In addition, the tropical Atlantic ozone burden was more strongly influenced by $NO_X$ from Africa than from South America. However, most of IAGOS data is acquired in the upper troposphere/lower stratosphere (UTLS) when the aircraft attain cruising altitude in the altitude band of 9–13 km. Thus, the ozone distribution in the middle and lower troposphere is still not well documented.

Satellite observations offer a great potential to overcome the limited spatial coverage of ground-based measurements and to fill the observational gap of air pollution. However, standard single-band ozone retrievals are not able to provide quantitative measurements of ozone abundance within the planetary boundary layer. Ultraviolet (UV) spaceborne spectrometers, like GOME-2 (Global Ozone Monitoring Experiment-2), have been used to observe tropospheric ozone with maximum sensitivity



at about 5–6 km altitude (e.g., Liu et al., 2010; Cai et al., 2012). Space-based thermal infrared (TIR) instruments, such as the Infrared Atmospheric Sounding Interferometer (IASI) on board the MetOp satellites, have shown good performance for observing ozone in the lower troposphere but with sensitivity peaking at lowest at 3 km altitude (e.g., Eremenko et al., 2008; Dufour et al., 2012). More recently, synergetic approaches using UV and TIR radiances simultaneously have been developed to improve the sensitivity to lower tropospheric ozone (e.g., Cuesta et al., 2013; Fu et al., 2018; Colombi et al., 2021). A

multispectral approach called IASI+GOME2, combining IASI observations in the TIR and GOME-2 measurements in the UV, shows remarkable skills in observing the horizontal distribution of ozone concentrations in the lowermost troposphere (LMT - here after defined as the atmospheric partial column below 3 km of altitude, Cuesta et al., 2013). Air-quality-relevant capabilities of IASI+GOME2 have been demonstrated by quantitatively describing the transport pathways, the daily evolution, and photochemical production in the lowermost troposphere during major outbreaks across east Asia (Cuesta et al., 2018) and

Europe (Cuesta et al., 2013; 2022; Okamoto et al., 2023).

With increases in computing performance and availability of satellite observations of trace gases, data assimilation has been applied with success in monitoring air quality (e.g., Flemming et al., 2015; Gelaro et al., 2017; Inness et al., 2019; Miyazaki et al., 2015; 2020). Data assimilation is a methodology that allows a physical-chemically based interpolation to fill in of the observational information gaps using a model and to provide an estimate of the most likely state and its uncertainty.

Applications of data assimilation to atmospheric chemistry can improve analyses of tropospheric pollution, and can provide estimates of tropospheric emissions (Lahoz and Schneider, 2014). The result of data assimilation is termed a "reanalysis" when the data assimilation approach is performed for past data by using a consistent system. There have been several studies that compared atmospheric chemistry reanalysis products. Air pollutants including ozone and CO derived from some chemistry reanalysis products were evaluated on regional scale in East Asia (Park et al., 2020; Ryu and Min, 2021; Zhang et al., 2022)

and Europe (Falk et al., 2021; Lacima et al., 2023). Huijnen et al. (2020) intercompared four atmospheric chemistry reanalysis products and reported that the standard deviation (SD) is the largest over South America, the Tropical Atlantic and Central Africa because of their differences of the representations of biomass-burning emissions and its impacts on ozone production, the representation of convective transport, and large uncertainties in biogenic emissions.

In the present paper, we intercompare ozone distributions of the multispectral satellite approach IASI+GOME2, in situ airborne

measurements conducted within the Atmospheric Tomography Mission (ATom) and three tropospheric chemistry reanalysis products over the Tropical and South Atlantic in February 2017. Section 2 describes the satellite data, in situ observations and atmospheric chemistry reanalysis products used for the analysis. Results and discussions on the distribution of tropospheric ozone and CO over the Tropical and South Atlantic is presented in Sect. 3. Conclusions are given in the last section.



## 2 Data and methods

In this study, we characterize the tropospheric ozone distribution over the Tropical Atlantic using data from a satellite approach and three chemistry reanalysis products. To analyse the origin of ozone plumes, CO satellite retrievals and three chemistry reanalysis products are also employed. We consider the region covering the Atlantic Ocean between 40° S and 40° N to investigate interactions of pollution and the transport between the tropics and the subtropics.

### 2.1 Satellite data

#### 2.1.1 IASI+GOME2 ozone multispectral observations

The multispectral satellite approach IASI+GOME2 is designed for observing lowermost tropospheric ozone by synergism of TIR atmospheric radiances observed by IASI and UV earth reflectances measured by GOME-2 (Cuesta et al., 2013; 2018). Both instruments are onboard the MetOp satellite series, and they both offer global coverage every day (around 09:00 local time) with a relatively fine ground resolution (12 km diameter pixels spaced by 25 km for IASI at nadir and ground pixels of 80 km × 40 km for GOME-2). Spectra and Jacobians in the IR and UV are simulated by the KOPRA (Karlsruhe Optimized and Precise Radiative transfer Algorithm; Stiller et al., 2002) and VLIDORT (Vector Linearized Discrete Ordinate Radiative Transfer; Spurr, 2006) radiative transfer codes, respectively. Ozone profiles are retrieved at the vertical grid between the surface and 60 km of altitude. The IASI+GOME2/MetOp-B ozone product including vertical profiles of ozone, averaging kernel, error estimations and quality flags is publicly available on the French data centre AERIS (https://iasi.aeris-data.fr, last access: November 2024). For reducing random errors, the product is averaged over a regular horizontal grid of 1° × 1°, in the same way as done by Cuesta et al. (2018). Ozone concentrations are provided as an average ozone volume mixing ratio in ppb within the layer, which is calculated as the ratio of each partial column of ozone and air.

#### 2.1.2 IASI carbon monoxide retrievals

The CO retrievals used in this study are derived from IASI radiances using the FORLI algorithm (Fast Optimal Retrievals on Layers for IASI; Hurtmans et al., 2012), from the Université Libre de Bruxelles (ULB) and the Laboratoire Atmosphères, Milieux, Observations Spatiales (LATMOS). This approach uses pre-calculated lookup tables of absorbance cross sections at various pressures and temperatures, and an optimal estimation method for the inverse scheme. The algorithm derives vertical profiles of CO, on a grid of 18 equidistant layers of 1 km of depth from the surface up to 18 km, and a unique layer from 18 to 60 km. The IASI/MetOp-B CO product including total and partial columns of CO derived by profile integrations, averaging kernels, error estimations and quality flags is publicly available on the French data centre AERIS (https://iasi.aeris-data.fr, last access: November 2024). The product is averaged over a regular horizontal grid of 1° × 1° and CO concentrations are calculated as an average CO volume mixing ratio in ppb within the layer in the same way as far IASI+GOME2 ozone.





## 2.2 Atmospheric chemistry reanalyses

The global atmospheric chemistry reanalysis products compared in this paper, the Tropospheric Chemistry Reanalysis version
2 (TCR-2), the Copernicus Atmosphere Monitoring Service reanalysis (CAMS reanalysis) and the Modern-Era Retrospective
Analysis for Research and Applications version 2 (MERRA-2), are listed in Table 1. The general configuration of the various
data assimilation systems is provided in the following subsections. We regrid these atmospheric chemistry reanalysis products
to $1° \times 1°$ resolutions for consistency with gridded IASI+GOME2 ozone data. Then, we convert their pressure levels to altitude
by using geopotential fields from ERA5 (Sect. 2.4).

**Table 1: Overview of the global atmospheric chemistry reanalysis products in 2017.**

|  | TCR-2 | CAMS reanalysis | MERRA-2 |
|---|---|---|---|
| Available period | 2005–2021 | 2003–present | 1980–present |
| Spatial resolution | $1.1° \times 1.1°$ | $0.7° \times 0.7°$ | $0.5° \times 0.625°$ |
| Vertical levels (top pressure level) | 32 layers (4.4 hPa) | 60 layers (0.1 hPa) | 72 layers (0.01 hPa) |
| Output frequency | 2 h (surface) 6 h (3D) | 3 h | 1 h |
| Meteorological fields | ERA-Interim | ERA-5 | GEOS-5 |
| Anthropogenic emissions | HTAPv2 | MACCity | AeroCom Phase II, EDGERv4.2 |
| Biomass burning emissions | GFEDv4 | GFASv1.2 | QFED |
| Biogenic emissions | Guenther et al. (2006) | Calculated by the MEGAN | Guenther et al. (1995) |
| Data assimilation scheme | EnKF | 4D-Var | 3D-Var |
| Assimilated ozone retrievals for 2017 | TES, MLS | MLS, OMI, GOME-2, SBUV/2 | OMI, MLS |
| Assimilated CO retrievals for 2017 | MOPITT | MOPITT | — |

## 2.2.1 Tropospheric Chemistry Reanalysis version 2 (TCR-2)

TCR-2) (Miyazaki et al., 2019) is a global atmospheric chemistry reanalysis based on the MIROC-CHASER (Model for
Interdisciplinary Research on Climate-Chemical atmospheric general circulation model for study of atmospheric environment
and radiative forcing, Watanabe et al., 2011) by the National Aeronautics and Space Administration (NASA) Jet Propulsion



Laboratory (JPL) (Miyazaki et al., 2020). TCR-2 uses an ensemble Kalman filter (EnKF) data assimilation technique to combine satellite observations of ozone, CO, nitrogen dioxide ($NO_2$), nitric acid ($HNO_3$) and sulphur dioxide ($SO_2$). TCR-2 has a T106 horizontal resolution ($0.7° \times 0.7°$) with 32 vertical levels from surface to 4.4 hPa. TCR-2 is available at 2-hour intervals for surface concentrations and at 6-hour intervals for 3-D concentrations. Meteorological fields used by TCR-2 are nudged towards the 6-hourly ERA-Interim (Dee et al., 2011). A priori surface emissions from anthropogenic sources are obtained from the HTAP version 2 for 2010 (Janssens-Maenhout et al., 2015). For biomass burning emissions, the monthly Global Fire Emissions Database (GFED) version 4 (Randerson et al., 2018) are used. Emissions from soils are based on monthly mean Global Emissions Inventory Activity (GEIA) (Yienger and Levy, 1995). Biogenic emissions from vegetation are considered for non-methane hydrocarbons (NMHCs) based on Guenther et al. (2006). In the present study, we use 2-hourly ozone and CO TCR-2 data (Miyazaki, personal communication, 2020).

**2.2.2 Copernicus Atmosphere Monitoring Service reanalysis (CAMSRA)**

CAMS reanalysis is a global atmospheric chemistry reanalysis based on the Integrated Forecast System (IFS) cycle 42R1 by the European Centre for Medium-Range Weather Forecasts (ECMWF) (Inness et al., 2019, last access: 12 April 2022). CAMS reanalysis uses the four-dimensional variational (4D-Var) data assimilation technique to combine satellite observations of ozone, CO, $NO_2$ and aerosol optical depth (AOD). The spatial resolution of CAMS reanalysis is a reduced Gaussian grid at a spectral truncation of T255, which is equivalent to grid spacing of approximately 80 km globally ($0.7° \times 0.7°$), with 60 vertical levels from surface to 0.1 hPa. CAMS reanalysis is available at 3-hour intervals. Daily global biomass burning emissions are provided by the Global Fire Assimilation System (GFAS) version 1.2 (Kaiser et al., 2012). Anthropogenic emissions are from the MACCity inventory (Granier et al., 2011), with modifications to increase wintertime road traffic emissions over North America and Europe following the correction of Stein et al. (2014). Monthly mean biogenic emissions are calculated offline by the Model of Emissions of Gases and Aerosols from Nature (MEGAN, Guenther et al., 2006) that used meteorological fields from the MERRA-2 following Sindelarova et al. (2014). Natural emissions from soils and oceans are taken from the Precursors of ozone and their effects in the Troposphere (POET) database (Granier et al., 2005; Olivier et al., 2003).

**2.2.3 Modern-Era Retrospective Analysis for Research and Applications version 2 (MERRA-2)**

MERRA-2 is a global atmospheric chemistry reanalysis based on the Goddard Earth Observing System version 5 (GEOS-5) atmospheric model, which is interactively coupled to the Goddard Chemistry Aerosol Radiation and Transport (GOCART) module (Chin et al., 2002), by NASA Global Modeling and Assimilation Office (GMAO) (Gelaro et al., 2017, last access: 2 March 2023). MERRA-2 uses three-dimensional variational data analysis (3D-Var) Gridpoint Statistical Interpolation (GSI) (Randles et al., 2017) to combine satellite observations of ozone, AOD and several meteorological fields. We utilize MERRA-2 CO product although it is not assimilated unlike TCR-2 and CAMS reanalysis but rather simulated by GEOS-5 modeling system. MERRA-2 uses cubed-sphere horizontal discretization at an approximate resolution of $0.5° \times 0.625°$ with 72 vertical



levels from the surface to 0.01 hPa. MERRA-2 is available at 1-hour intervals. For biomass burning emissions, the Quick Fire Emissions Dataset (QFED; Darmenov and da Silva, 2015) version 2.4-r6 is employed. Anthropogenic emissions are obtained

from AEROsol COMparisons between Observations and Models (AeroCom) Phase II (HCA0 v1; Diehl et al., 2012) and the Emissions Database for Global Atmospheric Research (EDGAR) v4.2, developed by the European Commission (Janssens-Maenhout et al., 2013). A monthly-mean varying climatology of terpene emissions was used (Guenther et al., 1995).

The MERRA-2 ozone profile product is recommended to use in the upper troposphere and the stratosphere (Bosilovich et al., 2015) because the use of a simplified chemistry and lack of emissions generally lead to an underestimate in ozone in the middle

to lower troposphere (Wargan et al., 2015). Knowland et al. (2017) demonstrated the capabilities of MERRA-2 when it is applied to the evaluation of stratospheric intrusions that influence on surface ozone air quality.

## 2.3 Observations used for evaluation

### 2.3.1 The Atmospheric Tomography Mission (ATom)

The Atmospheric Tomography Mission (ATom) is a NASA Earth Venture airborne field campaign to study the impacts of

175 human-produced air pollution on greenhouse gases and on chemically reactive gases over the Pacific and Atlantic oceans along a global-scale circuit (Thompson et al., 2022). ATom consists of four series of flights from ~82° N to ~86° S by using the long-range NASA DC-8 research aircraft. During these flights, the DC-8 repeatedly ascended and descended between ~0.2 and ~13 km in altitude. The four ATom circuits occurred in July–August 2016 (ATom-1), January–February 2017 (ATom-2), September–October 2017 (ATom-3), and April–May 2018 (ATom-4).

The ATom dataset includes merged data from all instruments (Wofsy et al., 2018) provided by the Oak Ridge National Laboratory Distributed Active Archive Center (ORNL DAAC, last access: 24 March 2023). We use ozone, CO, water vapor ($H_2O$), hydrogen cyanide (HCN), tetra chloroethylene ($C_2Cl_4$) and $NO_X$ observations.

We quantify the influence of biomass burning emission on the tropospheric ozone during ATom-2 according to the method of Bourgeois et al. (2021). The authors analyse in situ measurements of ozone ($O_3$) and $H_2O$, for defining air parcels with $O_3/H_2O$

$> 1$ ppbv ppmv$^{-1}$ and with $O_3/H_2O < 0.003$ ppbv ppmv$^{-1}$ as strongly influenced by stratospheric air and by marine air, respectively. To quantify the respective influence of biomass burning and urban emissions on each air parcel, they use a pair of HCN biomass of a burning tracer and $C_2Cl_4$ of an urban tracer. First, all sampled air masses are classified into four categories. For the region (40° S–40° N), air parcels are either defined as urban air (urban tracer > regional median, biomass burning tracer < regional median), biomass burning air (biomass burning tracer > regional median, urban tracer < regional median), mixed

pollution air (both urban and biomass burning tracers < regional median), and well-mixed and aged air (both urban and biomass burning tracers < regional median). The normalized excess mixing ratio (NEMR) of biomass burning and urban tracers is calculated according to following Eq. (1):

$$NEMR_X = \Delta X / \Delta CO, \tag{1}$$



where $NEMR_X$ is the normalized excess mixing ratio of compound X (i.e., X = HCN or $C_2Cl_4$) to CO, and $\Delta X$ ($\Delta CO$) is the difference between the mixing ratio of compound X (CO) and its background level. The background levels are defined as the average mixing ratio in well-mixed and aged air masses. The respective influence of urban and biomass burning emissions ($F_X$) is then calculated as the ratio of the NEMR of compound X to the average emission ratio of compound X ($ER_X$) as follows:

$$F_X = NEMR_X/ER_X, \tag{2}$$

Bourgeois et al. (2021) use 5.7 pptv ppbv$^{-1}$ and 0.03 pptv ppbv$^{-1}$ as ERs of biomass burning (HCN) and urban ($C_2Cl_4$) air according to Andreae (2019) and Kondo et al. (2004). Then, tropospheric ozone attributed to biomass burning and urban emissions are calculated as follows:

$$O_3^X = F_X \times \Delta O_3, \tag{3}$$

A dataset containing back trajectories and boundary layer influences of air parcels along the ATom flight tracks is also distributed by the ORNL DAAC (Ray, 2021, last access 22 December 2023). We use the back trajectories interpolated by using National Centers for Environmental Prediction (NCEP) Global Forecast System (GFS) meteorology. Model trajectories are initialized at receptors spaced 1 min apart along the ATom flight tracks, followed backwards for 30 days, and reported at 3-hour resolution. Boundary layer influences are determined based on 30-day back trajectories.

## 2.4 Meteorological data

Meteorological conditions leading to photochemical production of ozone and transport are described with the ERA5 reanalysis (Hersbach et al., 2020) produced by the European Centre for Medium-Range Weather Forecast (ECMWF). We use meteorological fields with global coverage, a horizontal resolution of $0.25° \times 0.25°$, 37 pressure levels, and a time step of 1 hour. Eastward and northward components of wind, vertical velocity, relative humidity, geopotential, and potential vorticity are used to describe transport patterns downloaded from the Climate Data Store (https://cds.climate.copernicus.eu/, last access: 19 December 2023).

The NO$_X$ production by lightning is generally represented by parameterizations in global chemistry transport models, resulting in differences between models and thus uncertainties. Scientific observations of lightning occurrence have been recorded from the space and the ground. The World Wide Lightning Location Network (WWLLN) is a global network monitoring lightning activity by very low frequency radio sensors (Dowden et al., 2002). Recently, a global, high-resolution gridded time series and climatology of lightning stroke density, the WWLLN Global Lightning Climatology (WGLC) has been published (Kaplan and Lau, 2021a) and is freely available at $0.5°$ and 5 arcmin spatial with daily and monthly temporal resolution (Kaplan and Lau, 2021b, last access: 19 April 2023).

As a convective proxy, we use monthly outgoing longwave radiation (OLR) data distributed by the National Oceanic and Atmospheric Administration (NOAA) Physical Science Laboratory (PSL) (Liebmann and Smith, 1996; https://psl.noaa.gov/, last access: 10 March 2023). The OLR is a good indicator of the position of the the Inter-Tropical Convergence Zone (ITCZ). Deep convective clouds present at the ITCZ are associated with OLR below 220 W m$^{-2}$ (e.g. Park et al., 2007).





## 2.5 Other data

The locations of fires are derived from the Terra and Aqua MODIS (Moderate Resolution Imaging Spectrometer) active fire products (MCD14ML Collection 6; Giglio et al., 2016) distributed by the Fire Information for Resources Management System (FIRMS; https://firms.modaps.eosdis.nasa.gov/, last access: 13 March 2020). This dataset provides the values of fire radiative power (FRP) and the inferred hotspot type: "presumed vegetation fire", "active volcano", "other static land source", and "offshore". We only use the FRP values of presumed vegetation fire with a confidence level greater than 50 %.


## 3 Results and discussion

The analysis of the distribution and origins of tropospheric ozone over the Tropical Atlantic is presented here in two steps. First, in Sect. 3.1, we analyse the monthly evolution of key elements that influence ozone distribution, namely tropical convective activity described by lightning and biomass burning emissions estimated by fire detections. We also quantify the differences between several tropospheric ozone products (from models and satellite observations) to identify the period with the largest uncertainties. This leads to a more detailed description of the tropospheric ozone distribution in one of the months with the largest uncertainties (February).


The following Sect. 3.2 looks in detail at the distribution of tropospheric ozone and carbon monoxide, an ozone precursor derived from combustion and thus used to identify the air mass origin of either biomass fires or anthropogenic activities. This is done with coincident in situ and satellite measurements and reanalysis during the period 13–15 February 2017. This section is followed by a detailed description of the origins of tropospheric ozone and its precursors, including an explanation of the gaps between observational (both in situ and satellite) and modelling (atmospheric chemistry reanalysis) data sets.


### 3.1 Monthly evolution of tropospheric ozone and related variables over the Tropical Atlantic

The climate in the tropics is characterised by alternating wet and dry seasons depending on the position of the ITCZ. It moves north in the Northern Hemisphere summer and south in the Northern Hemisphere winter. Lannuque et al. (2021) defined two main seasons (from December to March and from June to October) and two transition seasons (from April to May and November) in the African inter-tropical zone based on the position of ITCZ, defined by zonal and meridional winds and relative humidity because the classical four seasons were not adapted to their study area.


According to the maps presented in Figs. 1 and 2, lightning and fire activities can be observed throughout the year in both, the African and the South American continents, with large seasonal variability, clearly associated with the ITCZ annual shift. Figure 1 shows the distributions of monthly WGLC lightning density together with low values of OLR suggesting the location of deep convection activity in February, May, August, and November 2017. Estimations of the regional lightning source altitudes by the satellite (derived by Peterson 2022) suggest that the source altitude distributions peak at 10–11 km altitude




over the Amazon and the Congo Basins. In the La Plata Basin and South Africa, they peak at lower altitude. Therefore, wind vectors at 10 km of altitude are presented in Fig. 1. In February, the highest lightning densities are observed in the Gulf of Guinea, Central and Southern Africa, and over South America (Fig. 1a). This clearly shows that this is the location where strong ascending motions within the troposphere are expected, which may bring air pollutants emitted from the surface such as ozone precursors up to the middle and upper troposphere.

In May, convective activity shifts northwards in location as suggested by the decrease of lightning in South America and peaks over Colombia and Venezuela (Fig. 1b) and an increase over the African Sahel in a vast west-to-east band. While lightning density distribution in August is quite similar to the previous month, the density decreases in Western Africa (Fig. 1c). In November, the centre of lightning activities moves southwards as a consequence of the ITCZ shift (Fig. 1d). The highest lightning densities are detected in the Congo Basin, Brazil, Northern and Western South America. In this period (May–August),

deep convection likely affects atmospheric circulation mainly north of the equator.





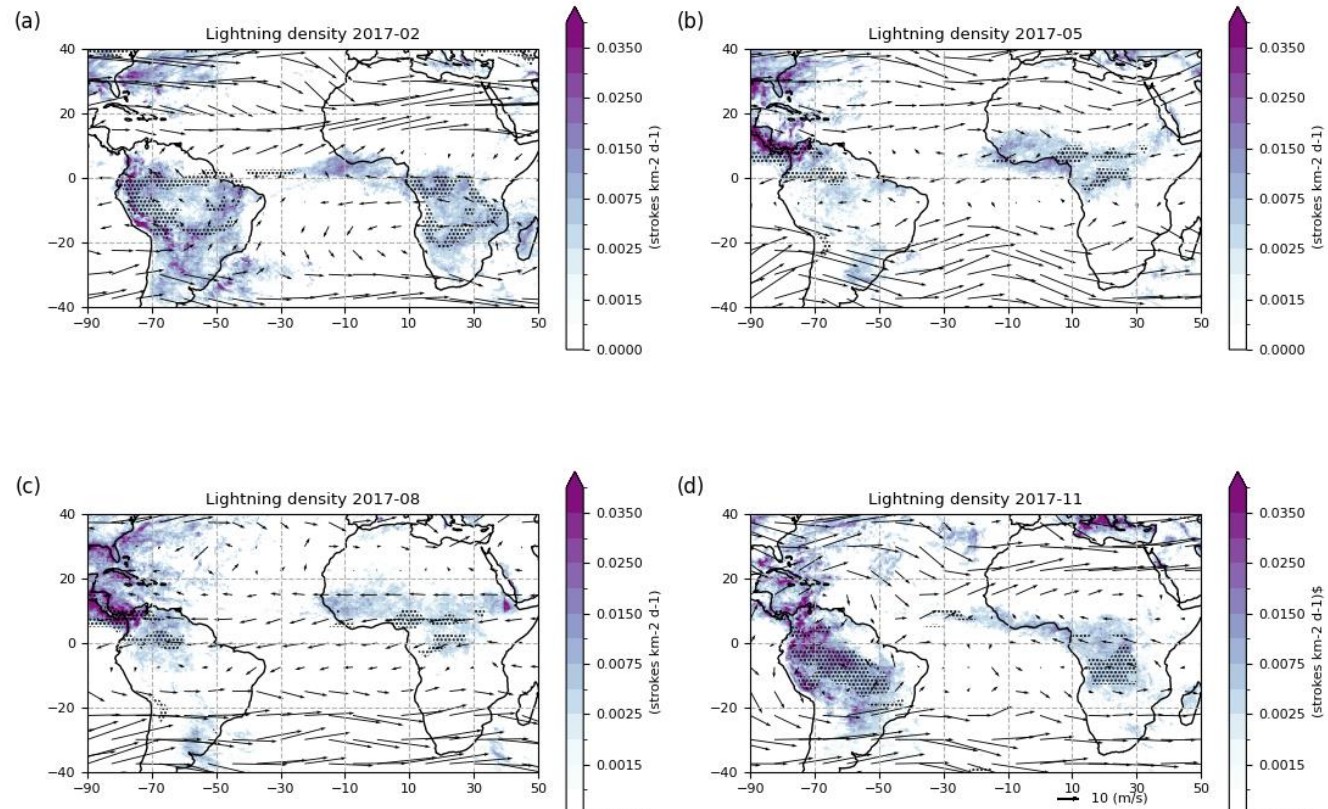

**Figure 1: Monthly WGLC lightning density in (a) February, (b) May, (c) August and (d) November 2017. Winds at 10 km of altitude from ERA5 are indicated by black arrows. Black dots indicate areas with OLR < 220 W m⁻².**


According to FRP maps presented in Figure 2, fire activity also varies with the ITCZ annual shift. In February, enhancement of FRP can be observed in the Caribbean North, Southern Cone, Western and Central Africa (Fig. 2a). Following the wind flow (shown at 3 km of altitude in Fig. 2), trace gases and smoke emitted by biomass fires are then expected to largely affect the Tropical Atlantic. In May, while the location of the highest FRP values is similar to previous months, the intensity decreases

in South America and the Sahel south of the Sahara Desert, and increases in the Congo Basin (Fig. 2b).

In August, the fire intensity is the highest over the Congo Basin and Brazil (Fig. 2c). In November, fires are still detected in South America and Southern Africa, and restarts over the Sahel south of the Sahara Desert (Fig. 2d). This is consistent with





previous studies suggesting that the peak of burning events over Africa north of the equator occurs in January, whilst south of

it in July (Roberts et al., 2009). May and October are expected to be the periods of transition between hemispheres. Consistency

is also found over South America, where biomass burning emissions start increasing in June, enhance in July and August, and

peak in September, and start decreasing in October (Pereira et al., 2022).

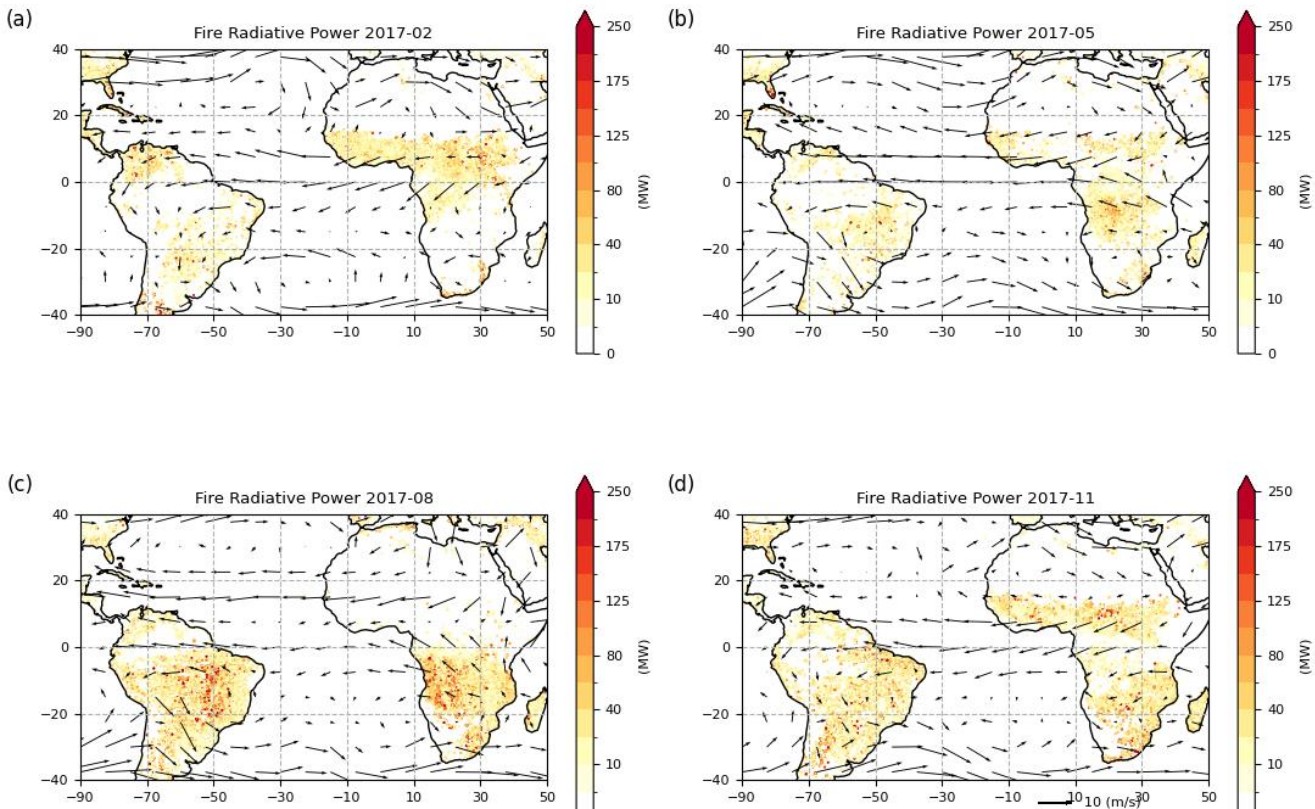

**Figure 2: Monthly fire radiative power (FRP) of presumed vegetation fire in (a) February, (b) May, (c) August and (d) November 2017. Winds at 3 km altitude from ERA5 are indicated by black arrows.**


There are strong seasonal variations in the location of fires and deep convection, and consequently, change in the regime of

atmospheric composition, for instance tropospheric ozone, is large over the Tropical Atlantic. Given these complex conditions

over this remote area, we start our analysis with an initial estimate of the uncertainties of atmospheric modelling of tropospheric



ozone as regional SD between several atmospheric chemistry reanalyses and satellite measurements. We consider monthly
averaged horizontal distributions of ozone in the lowermost troposphere (defined here as the atmospheric layer between the
surface and 3 km above sea level) over the Tropical Atlantic (25° S–25° N, 34° W–18° W). Figure 3 shows SD calculated
from three (reanalysis products only) or four (including IASI+GOME2) ozone products. Both monthly variations of SD show
clear seasonality with two maxima from December to February and from June to August, and with two minima from April to
May and from September to October. The maxima of differences correspond to the two biomass burning seasons over Africa
and the relative minima correspond to the two transition periods. The largest SD (up to ~8 ppb) between three and four products
can be observed from November to March, corresponding to the biomass burning season in the Northern Hemisphere (Fig. 2a)
and deep convection over Central Africa and the Gulf of Guinea (Fig. 1a).

A previous intercomparison of four chemistry reanalyses including TCR-2 and CAMS reanalysis, suggests the largest SD at
850 hPa over South America, Central Africa, and Northern Australia (Huijnen et al. 2020). This is mainly associated with the
300 differences of the representation of biomass burning emissions and its impact on ozone production among the systems.
Estimates of CO emissions for African fires have been subject to considerable uncertainty because of the high variability of
African fires in time and space (Andela and van der Werf, 2014). Three chemistry reanalysis products used in this study adopt
different biomass burning emission inventories (Table 1). GFED used by TCR-2 is produced by using the bottom-up approach
which uses burned area and fuel loads (van der Werf et al., 2017). GFAS (used by CAMS reanalysis) and QFED (used by
305 MERRA-2) are produced by using the top-down approach which uses FRP data (Darmenov and da Silva, 2015; Kaiser et al.,
2012). Mismatches between bottom-up and top-down approaches have been discussed in previous studies (e.g., Stoppiana et
al., 2010; van der Werf et al., 2006; Zheng et al., 2018). Evaluations of uncertainties of fire emissions and their impact on the
ozone production are beyond the scope of this study.





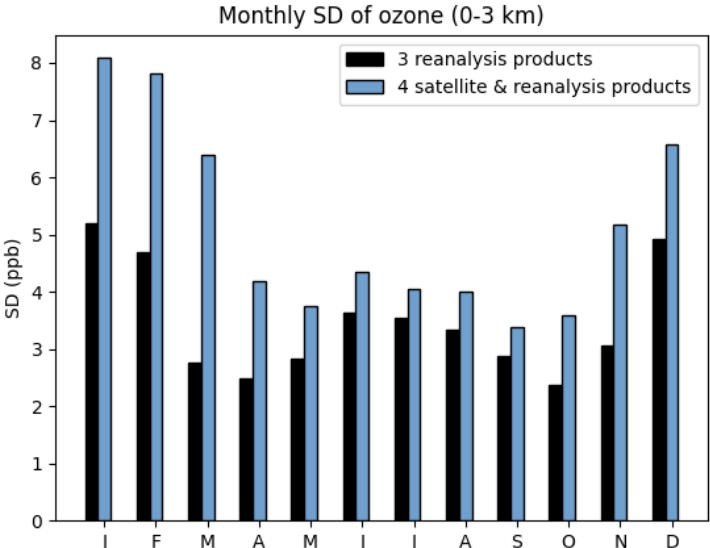

**Figure 3: Evaluation of regional SD of monthly mean ozone in the lowermost troposphere in 2017 over the Equatorial Atlantic Ocean (25° S–25° N, 34° W–18° W). Black and light blue bars indicate the SD calculated from three atmospheric chemistry reanalysis products (TCR-2, CAMS reanalysis and MERRA-2) and four products (IASI+GOME2 and three chemistry reanalysis products), respectively.**

Figure 4 shows horizontal distributions of monthly mean ozone and the SD in the lowermost troposphere in February 2017, within the period shown from Fig. 3 as that with the largest differences. The IASI+GOME2, TCR-2 and CAMS reanalysis products show high concentrations of ozone from Western Africa to the Gulf of Guinea (Fig. 4a–c). MERRA-2 does not depict any enhancement of ozone in Western Africa probably because of the use of its simplified chemistry and lack of emissions, as mentioned by Wargan et al. (2015). An enhancement of ozone in the north of the St. Helena anticyclone between 10° S and 20° S and centred at 20° W can be only observed by IASI+GOME2, whereas three chemistry reanalysis products show low ozone values below 20 ppb. TCR-2 and CAMS reanalysis show relatively higher concentration of ozone over the Atlantic in the Northern Hemisphere as compared to IASI+GOME2. The distribution of SD also reflects the above-mentioned features (Fig. 4e). The largest SD can be seen over Central Africa east of the Gulf of Guinea and also in the outflow over the Tropical Atlantic Ocean, which may be associated with differences of the representation of biomass burning emissions and its impacts on ozone production. Also, large SD can be seen in the north of the St. Helena anticyclone as ozone is only observed by IASI+GOME2. In addition, moderate SD can be also seen over the North Atlantic between 10° N and 40° N (Fig. 4e).



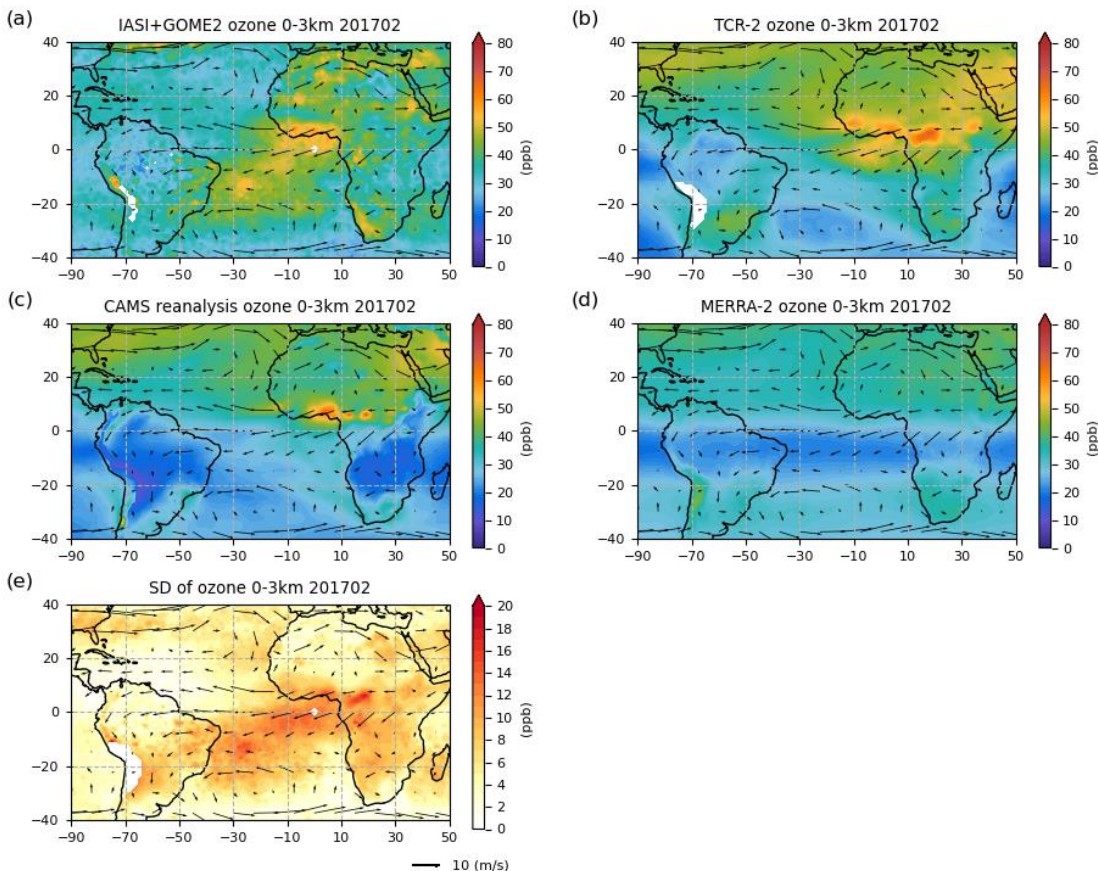

**Figure 4: Distribution of monthly mean ozone in the lowermost troposphere in February 2017. (a) IASI+GOME2, (b) TCR-2, (c) CAMS reanalysis, (d) MERRA-2, and (e) SD Winds at 3 km altitude from ERA5 are indicated by black arrows.**

A rather different situation is seen for ozone distribution depicted in the middle to upper troposphere. This is shown by Fig. 5 in terms of monthly mean ozone and the SD in the atmospheric layer between 6 km and 12 km above sea level, in February 2017. All products show a horseshoe-shaped structure of high concentration of ozone from Southern and Western Africa to the east of Brazil. In this upper layer of the troposphere, more similarity can be observed between the reanalyses (Fig. 5). It

might come from the influence of the assimilation of ozone satellite products. IASI+GOME2 show lower ozone concentration in South America as compared to three chemistry reanalysis products. The largest SD can be seen over the Atlantic north of the equator and South America in the Northern Hemisphere (Fig. 5e). Both the ozone and SD spatial distributions over the





Atlantic are clearly different in this upper tropospheric layer (Fig. 5) as compared to the lowermost troposphere (Fig. 4). The next section discusses in detail these differences and compares them with in situ reference measurements performed by an

aircraft during 13–15 February 2017.

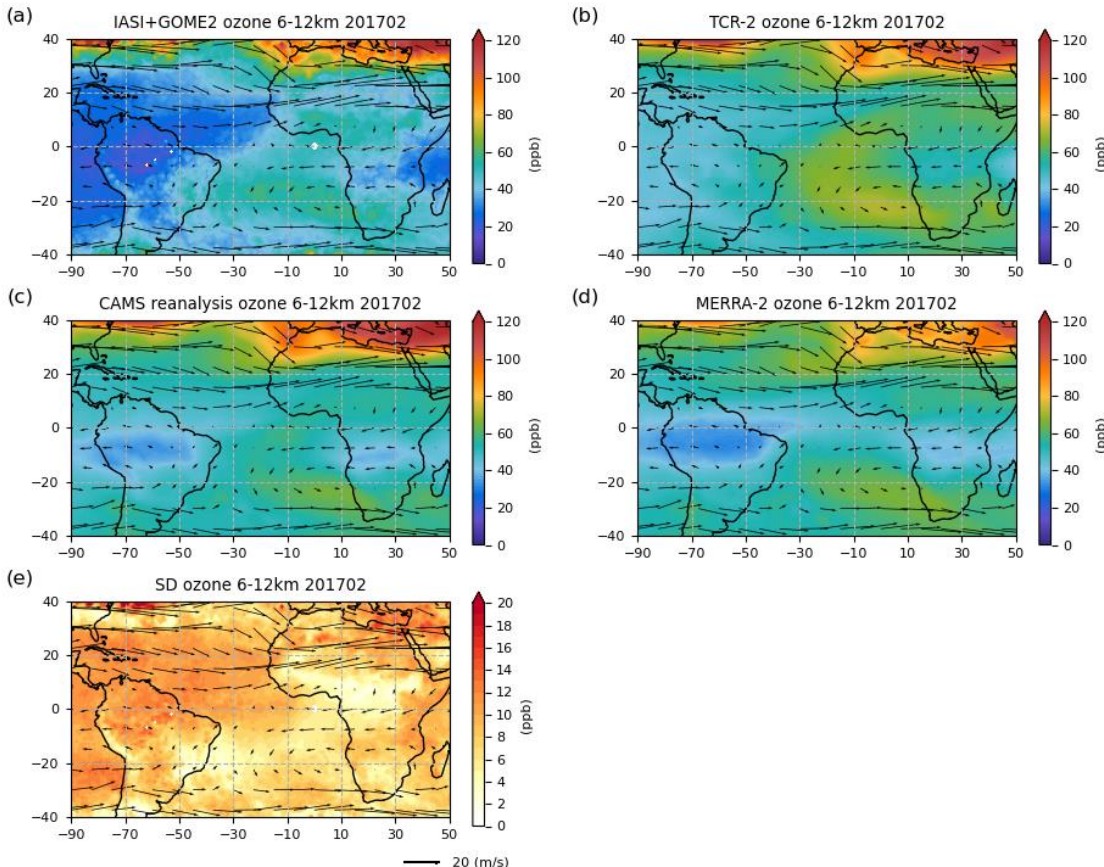

**Figure 5: Distribution of monthly mean ozone in the middle and upper troposphere in February 2017. (a) IASI+GOME2, (b) TCR-2, (c) CAMS reanalysis, (d) MERRA-2, and (e) SD Winds at 9 km altitude from ERA5 are indicated by black arrows.**





**3.2 Case study over the Atlantic during 13–15 February 2017**

This subsection provides a detailed analysis of this case study, in terms of tropospheric ozone (Sect. 3.2.1) and carbon monoxide (Sect. 3.2.2) spatial distributions. Then, we use the in situ measurements of chemical tracers, together with satellite measurements, for describing the origins of tropospheric ozone over the Atlantic (Sect. 3.2.3).

**3.2.1 Tropospheric ozone spatial distribution**

In order to better understand the differences between atmospheric chemistry reanalyses and satellite observations, we focus here on the period and location of ATom-2 in situ observations of tropospheric ozone and CO in February 2017. The NASA DC-8 aircraft transect over the Atlantic during 13 and 15 February 2017 covers from south to north, respectively, from 53° S to 8° S and from 8° S to 39° N (on each of these days, see the flight track in Fig. 6). Therefore, we consider here concentrations of the four products (IASI+GOME2, TCR-2, CAMS reanalysis and MERRA-2) averaged for the period from 13 to 15 February 2017.

Figure 6 shows horizontal distributions of ozone in the lowermost troposphere for this period. These ozone distributions are generally similar to the monthly mean distributions in Fig. 4a–d. Except for MERRA-2, all other products show enhancement of ozone concentrations over active biomass burning areas near the coast of the Gulf of Guinea and over the nearby Sea following the wind flow. Only IASI+GOME2 shows high ozone concentration (> 50 ppb) in the north of St. Helena anticyclone (centred at 30° S, 40° W). TCR-2 and CAMS reanalysis show relatively higher concentration in the Northern Hemisphere as compared to IASI+GOME2. The overall similarities between the ozone concentrations during 13–15 February 2017 and the average over the whole month indicate that the aspects studied in the 3-day period, such as the origins of tropospheric ozone (Sect. 3.2.3), are likely stable features representative of a larger period.





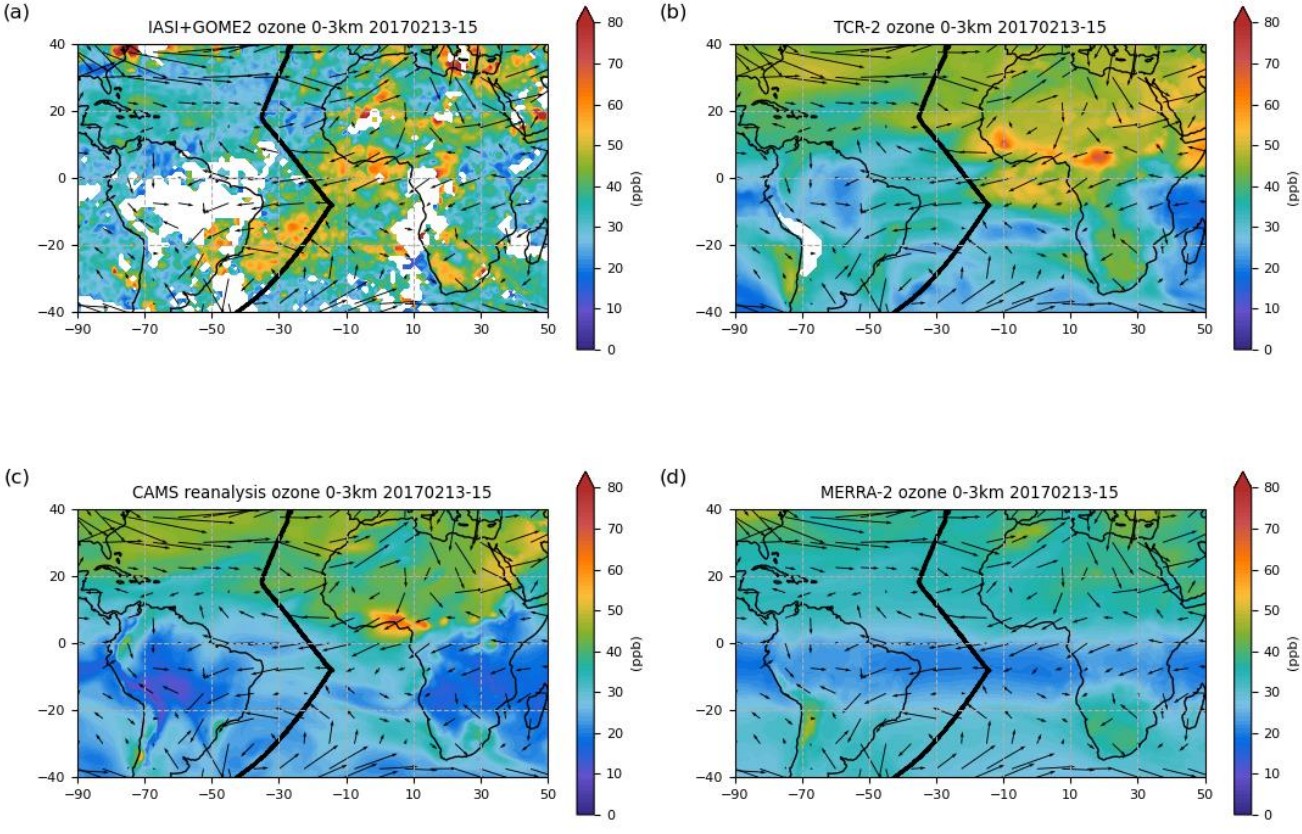

**Figure 6: Distribution of mean ozone in the lowermost troposphere during 13–15 February 2017. Black bold lines in (a) IASI+GOME2, (b) TCR-2, (c) CAMS reanalysis and (d) MERRA-2 indicate the ATom-2 flight track. Winds at 3 km altitude from ERA5 are indicated by black arrows.**

Middle-upper troposphere ozone (from 6 to 12 km of altitude) horizontal distributions are shown in Figure 7 for the period of 13–15 February 2017. As at the lowermost troposphere, similar distributions are found with respect to monthly averages in Fig. 5a–d. Three reanalyses show relatively higher concentration of ozone over the Atlantic in the Northern Hemisphere as compared with IASI+GOME2. All products show the previously remarked ozone plume forming a horseshoe-shape from Southern Africa to the east of Brazil (20° S) and until the Gulf of Guinea. Especially, ozone concentrations from TCR-2 are





the highest ones over the South Atlantic (about 80 ppb). MERRA-2 shows similar distribution with the other reanalysis
products unlike in the lowermost troposphere (Fig. 6).

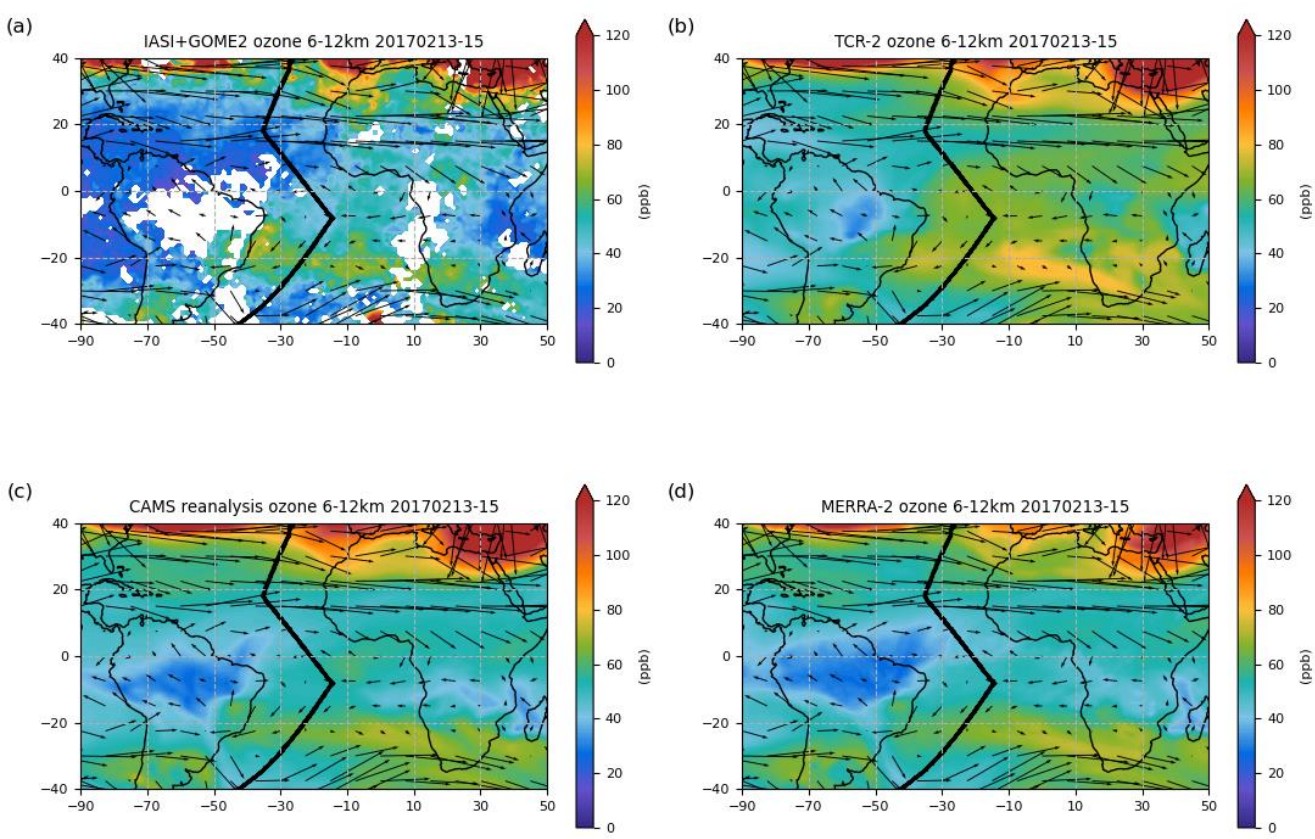

**Figure 7: Distribution of mean ozone in the middle and upper troposphere during 13–15 February 2017. Black bold lines in (a)
IASI+GOME2, (b) TCR-2, (c) CAMS reanalysis and (d) MERRA-2 indicate the ATom-2 flight track. Winds at 9 km altitude from
ERA5 are indicated by black arrows.**


To assess the accuracy of the satellite and reanalysis ozone products, we compare the vertical profiles along the flight track of
ATom-2. Figures 8 shows transects of vertical profiles of ozone concentration along the flight track of ATom-2 indicated by
bold black line in Figs 6 and 7. The ozone distribution of four products is characterized by an ozone plume in the troposphere
between 30° S and 5° N below 10 km of altitude. IASI+GOME2 depicts the ozone plume at 3–6 km, which is lower than the



altitude of the plume shown by the reanalysis products (Fig. 8). This enhancement of ozone concentration in IASI+GOME2 corresponds the one seen in the lower troposphere horizontal distribution over the South and Tropical Atlantic (Fig. 6a). The horizontal distribution of ozone from the reanalyses presents lower concentrations in the lowermost troposphere between 30° S and 5° N (Fig. 6b–d), as the ozone plume is located at higher altitude (> 5 km) in this case (Fig. 8b–d). Particularly in the Northern Hemisphere (north of 10° N), we can observe large difference of ozone concentration between IASI+GOME2 and

three reanalyses (Fig. 8). This difference corresponds to concentrations 20 ppb lower for IASI+GOME2 than for the reanalyses. In addition, an enhancement of ozone from the upper troposphere to the middle and lower troposphere can be observed in reanalyses around 25° N. However, IASI+GOME2 shows only an enhancement of ozone concentrations up to 40 ppb in the lower troposphere around 15° N. The satellite-derived vertical profiles do not show larger values until the lower stratosphere.



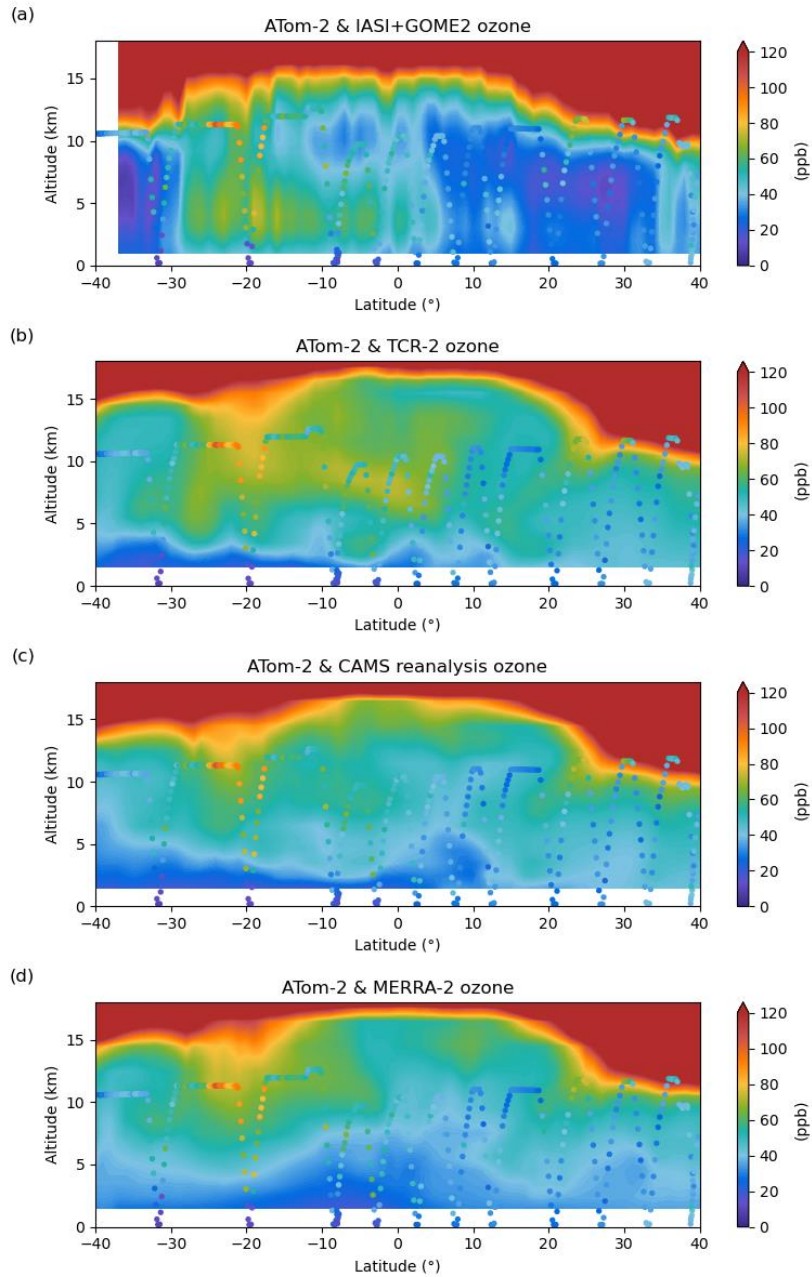

**Figure 8: Vertical profiles of ozone concentrations for the period from 13 to 15 February 2017. Dots indicate ATom-2 observations. The ozone concentrations of (a) IASI+GOME2, (b) TCR-2, (c) CAMS reanalysis and (d) MERRA-2 are averaged for the period from 13 to 15 February 2017, and are extracted along the ATom-2 flight track.**





The transect of ATom-2 in situ measurements and IASI+GOME2 shows a remarkable agreement, across the whole south-north track and within the whole troposphere (Fig. 8a). This agreement is clearly better than with respect to three reanalyses. Both ATom-2 and IASI+GOME2 depict a similar structure of the ozone plume in the troposphere (2 to 7 km of altitude) located between 30° S to 5° N. At about 3 km, ATom-2 ozone concentrations are 68 ppb (20.1° S) and 54 ppb (19.1 °S), near the values depicted by IASI+GOME2 (59–70 ppb between 19° S to 21° S). Lower concentrations are shown by TCR-2, CAMS

reanalysis and MERRA-2, respectively of 44–51 ppb, 34–40 ppb and 36–37 ppb. This region is corresponding to the region where only IASI+GOME2 shows high concentration in the north of St. Helena anticyclone (Figs. 4 and 6). It suggests that three reanalysis products underestimate the ozone concentration in the lower troposphere at around 20 °S. In the in the middle troposphere (4–7 km), ATom-2 measure similar ozone levels as both IASI+GOME2 and TCR-2. On the other hand, Atom-2 shows larger concentrations in the upper troposphere at 11 km of altitude (> 90 ppb) than the satellite and the three reanalysis

products.

In the Northern Hemisphere (north of 5° N and until 35° N), a clear decrease of ozone concentrations within the troposphere (down to about 30 ppb) is clearly observed by both Atom-2 and IASI+GOME2 (Fig. 8a). On other hand, none of three reanalyses depict such ozone reduction (Fig. 8b–d). A quantitative assessment of the difference to evaluate their capability is presented in Table 2. We compare ozone concentrations of four products (IASI+GOME2 and three reanalyses) with ATom-2

between 10° N and 30° N. In the lowermost troposphere, IASI+GOME2 shows lower mean concentration (24.8 ppb) compared to ATom-2 (30.7 ppb), whereas the other reanalysis products show much higher concentrations. Mean ozone concentrations from TCR-2 and CAMS reanalysis are 10–12 ppb higher. We only find a robust correlation between IASI+GOME2 and ATom-2 ozone concentrations (R = 0.58, p-value < 0.05). In the middle to upper troposphere, all products show robust correlations with respect to ATom-2 dataset (between 0.61 to 0.73), although only IASI+GOME2 show similar average concentrations

(39.1 and 42.5 ppb for the in situ and satellite measurements respectively). Reanalyses tend to overestimate these upper troposphere concentrations (by 16–20 ppb). These results suggest that the ozone concentration over the Atlantic in the Northern Hemisphere is overestimated by three chemistry reanalyses both in the lowermost troposphere and in the middle and upper troposphere, while the best agreement with ATom-2 is clearly seen for IASI+GOME2 in correlation and absolute values.



**Table 2: Summary ozone statistics for the lowermost troposphere (0–3 km) and the middle and upper troposphere (6–12 km) along the ATom-2 flight track between 10°N and 30°N. R is correlation coefficient with respect to ATom-2. Asterisk denotes statistical significance at p-value < 0.05 (*) and p < 0.01 (**).**



|  | ATom-2 | IASI+GOME2 | TCR-2 | CAMS reanalysis | MERRA-2 |
|---|---|---|---|---|---|
| 0–3 km (number of data = 13) | | | | | |
| Mean | 30.7 | 24.8 | 43.5 | 41.0 | 36.0 |
| Median | 31.6 | 24.6 | 44.5 | 41.1 | 35.4 |
| SD | 4.3 | 4.4 | 6.3 | 4.2 | 3.5 |
| R |  | 0.58* | –0.24 | 0.26 | 0.07 |
| 6–12 km (number of data = 66) | | | | | |
| Mean | 39.1 | 42.5 | 59.8 | 56.0 | 55.6 |
| Median | 32.4 | 34.0 | 53.0 | 49.8 | 51.7 |
| SD | 14.4 | 26.6 | 14.4 | 16.9 | 12.7 |
| R |  | 0.73** | 0.66** | 0.61** | 0.71** |

### 3.2.2 Carbon monoxide concentrations in the troposphere

Tropospheric ozone is a secondary pollutant, which is both chemically produced and destroyed during the transport in the atmosphere. Therefore, understanding its origin is a complex task. A first analysis of the origin of air masses rich in tropospheric ozone is presented here by investigating the vertical profiles of CO concentrations in a similar way as done for ozone in the previous subsection (Sect. 3.2.1). Figures 9 shows vertical profiles of CO concentration along the flight track of ATom-2 (indicated in Figs 6 and 7). They correspond to satellite retrievals of CO derived from IASI measurements and CO concentrations derived from the three reanalyses. The four products are characterized by a maximum of CO abundance around the equator in the lowermost troposphere (Fig. 9). This is consistent with Atom-2 measurements of CO. The CO concentrations retrieved by IASI are in good agreement with those measured by ATom-2 (> 250 ppb), which are clearly higher than the CO abundance in the reanalysis products. At about 5° S, the CO maximum underestimated by the three reanalyses is located at 2–3 km of altitude. It is composed of three CO plumes: from surface to the middle troposphere (15° S–15° N), from the middle troposphere to the upper troposphere (25° S–5° N) and in the upper troposphere (10° S–20° N). ATom-2 measurements are consistent with these CO enhancements in the upper and middle troposphere. In addition, all products and ATom-2 measurements show an enhancement of CO in the Northern Hemisphere (north of 20° N).



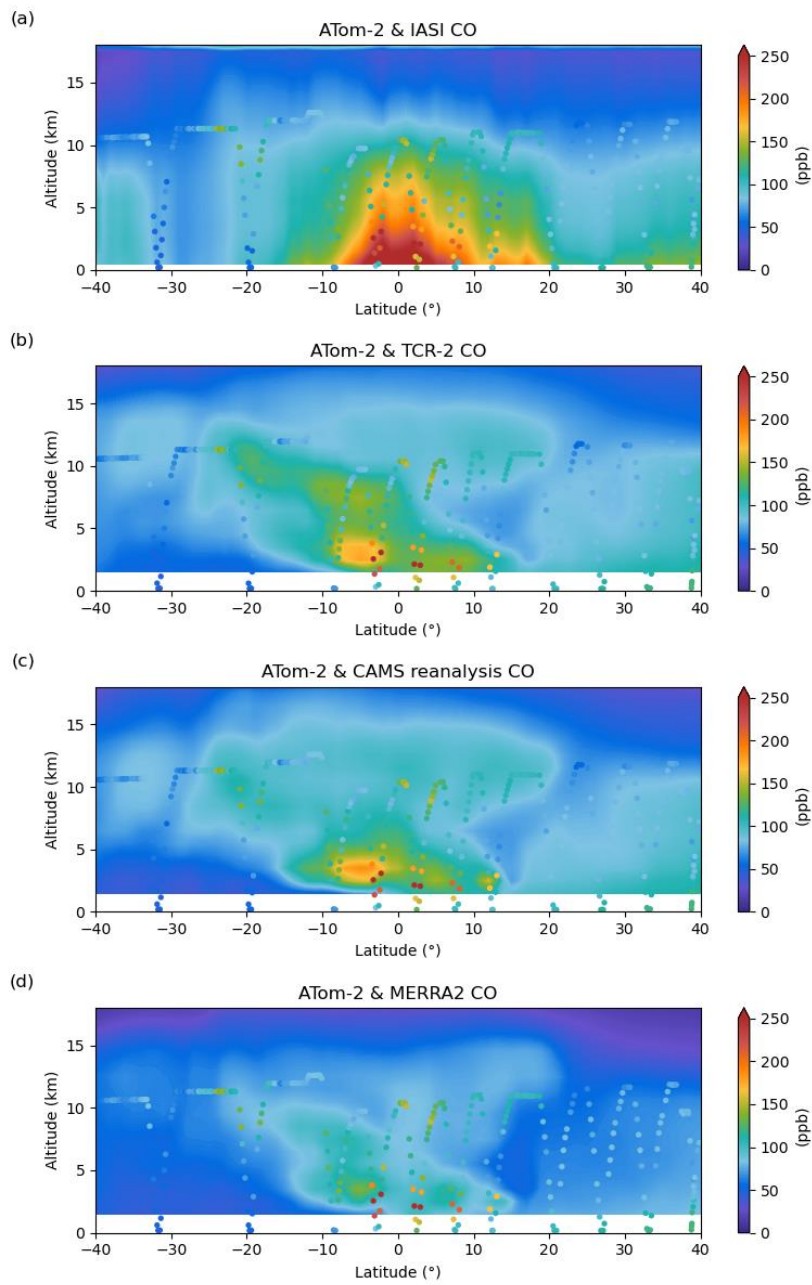


**Figure 9: Vertical profiles of CO concentrations for the period from 13 to 15 February 2017. Dots indicate ATom-2 observations. The CO concentrations of (a) IASI, (b) TCR-2, (c) CAMS reanalysis, and (d) MERRA-2 are averaged for the period from 13 to 15 February 2017, and are extracted along the ATom-2 flight track.**




Further insights of the consistence between ATom-2 and the satellite and reanalyses datasets can be analysed from ratios between ozone and CO concentrations ($O_3$/CO). Enhancements of this ratio along transport suggest ozone photoproduction (e.g. Cuesta et al., 2018), while it may vary between different air masses depending on their origin or other processes affecting its chemical composition. Figures 10 shows vertical profiles of ratio of ozone and CO concentrations along the flight track of ATom-2. Although the vertical sensitivity of IASI differs from that of IASI+GOME2, the corresponding $O_3$/CO ratio generally

shows a good agreement with that of ATom-2 all along the track, over the South, Tropical and North Atlantic between 30° S and 40° N (Fig. 10a). Both in situ measurements and satellite retrievals depict high values near 1 over the South Atlantic (30° S to 20° S) below 10 km of altitude. North of this area (5° S–35° N), lower values of the ratio between 0.2 and 0.4 are consistently shown by both datasets.

The enhancement of the $O_3$/CO ratio over the South Atlantic (25° S) is also present in the three reanalyses. MERRA-2 shows

higher ratio than the others. On the other hand, the three reanalyses display a second maximum around 20° N (Fig. 10b–d), which is not depicted by none of the observational datasets (satellite or in situ). This is collocated with a descending branch of a Hadley cell (see Sect. 3.2.3.1). Some punctual high values of the $O_3$/CO ratio measured by ATom-2 suggest the occurrence of a relatively small stratospheric instruction, contributing with ozone-rich air masses which are poor in CO concentrations. The reanalyses clearly overestimate the $O_3$/CO ratio in this area, likely linked to an overestimation of the magnitude and size

of the stratospheric air intrusion (which is not observed by IASI+GOME2) combined with underestimations of CO concentrations by the reanalyses (Fig. 9d).





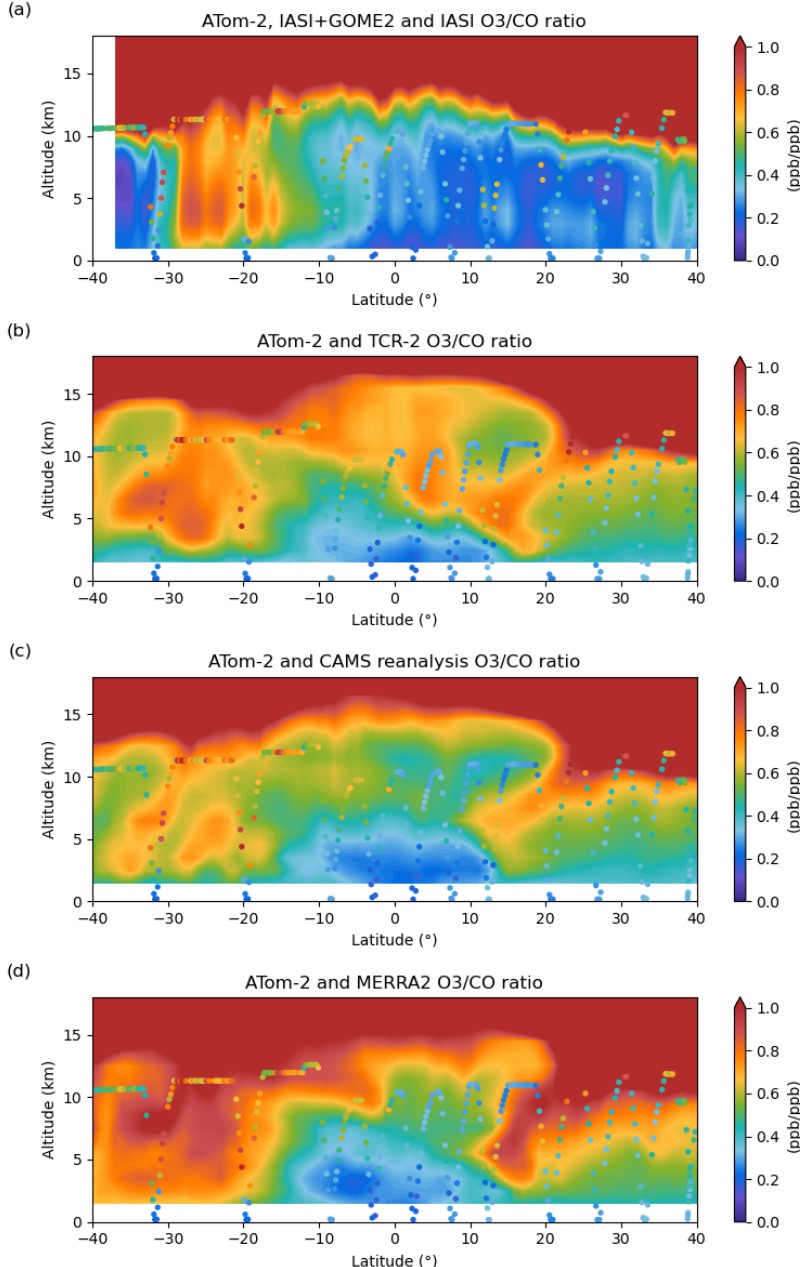

**Figure 10: Vertical profiles of the ratio of ozone and CO concentrations averaged over the period from 13 to 15 February 2017. Dots indicate ATom-2 observations. The ratios of IASI+GOME2 and (a) IASI, (b) TCR-2, (c) CAMS reanalysis, and (d) MERRA-2 are calculated by the average ozone and CO concentrations for the period from 13 to 15 February 2017, and are extracted along the ATom-2 flight track.**





### 3.2.3 Origins of ozone and CO reaching the Tropical Atlantic Ocean

Figure 11a shows a classification of multiple air masses (stratospheric air, marine air, urban air, biomass burning air, mixed
pollution air, and well-mixed and aged air) based on the method of Bourgeois et al. (2021). Here, well-mixed and aged air is
defined as air mass with simultaneous low levels of biomass burning (HCN) and urban ($C_2Cl_4$) tracers. The other significant
sources of CO (e.g., biogenic emissions and methane oxidation) and ozone precursors (e.g., lightning and soil emissions for
$NO_X$, biogenic emission of VOCs) are included in the well-mixed and aged air mass category. This classification provides a
very interesting picture of the complexity of the multiple contributions of the air masses along the transect. We clearly depict
a strong urban influence north of 10°N between 2 and 10 km of altitude, a sector with rather weak ozone abundance and low
$O_3$-to-CO ratio according to satellite/in situ measurements (Figs. 8a and 10a). Biomass burning emissions notably affect the
low-to-upper troposphere (from 2 to 11 km of altitude) roughly between 25° S to 5° N, which are collocated with moderate
and large abundances respectively ozone and carbon monoxide (Fig. 9a) much likely associated with emissions from Central
African fires. Marine and stratospheric influences can be seen along the transect near the ocean (below 2 km of altitude) and
close to the tropopause (above 10 km of altitude, specifically near 20°S and 25°N), respectively.





**Figure 11: Classification and origin of air masses along the transect sampled by ATom-2. (a) Air masses are classified into six categories: marine (navy), stratosphere (light blue), urban (light brown), biomass burning (dark green), mixed pollution (black),**




**and well-mixed and aged air (grey). Grey triangles indicate the well mixed and aged air influenced by lightning emissions. (b–g)**
**Colours indicate average time since trajectories are initialized with NCEP winds.**

To further analyse the chemical composition and origin of the air masses sampled along the ATom-2 transect, we illustrate the
correlative variation of NO$_X$ and HCN (the biomass burning tracer) concentrations measured by the aircraft (Fig. 12a). Marine,

urban, and mixed pollution air masses generally show low NO$_X$ concentrations. Stratosphere, biomass burning, and well-mixed
and aged air masses show higher NO$_X$ concentrations. There are two possible sources of NO$_X$ in well-mixed and aged air:
boundary layer (including biogenic and soil emissions) and upper troposphere (lightning emission). To distinguish these two
sources, we use probability of boundary layer influences determined by 30-day back trajectories (Ray, 2021; Fig. 12b). Well-
mixed and aged air masses with relatively high NO$_X$ concentration in Figure 12a show low probability of boundary layer

influences. Therefore, these air masses might be influenced by lightning emissions. We define air masses influenced by
lightning as those with high NO$_X$ concentration (> regional median 0.033 ppb) and low probability of boundary layer influences
(< 50 %). The significant influence of lightning is indicated by grey triangles in Figure 11a.

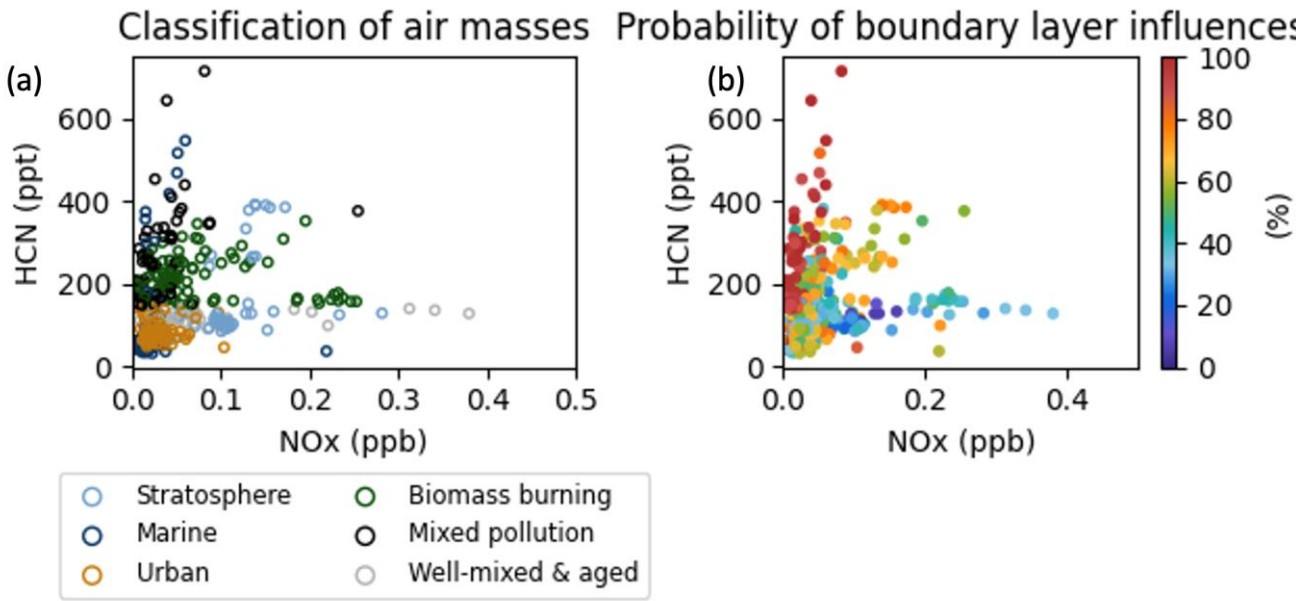

**Figure 12: Scatterplots of NO$_X$ vs. HCN (biomass burning tracer) of ATom-2 observation with colours indicating (a) the air masses**
**classification into six categories: marine (navy), stratosphere (light blue), urban (light brown), biomass burning (dark green), mixed**
**pollution (black), and well-mixed and aged air (grey). (b) the probability of boundary layer influences.**

The capability to identify the origin and nature of air masses by the satellite observations is illustrated in Fig. 13a is terms of
the relationship of the abundance of CO and ozone, coloured according to their origin (derived from ATom-2 measurements).



We remark a rather similar distribution of values as obtained for the scatter of values of HCN vs NOx in Fig. 12a. Urban-influenced air masses (yellow circles) are mostly associated with moderate abundances of both CO and ozone (up to respectively 150 and 60 ppb). Larger concentrations of ozone (> 70 ppb) retrieved by satellite correspond to stratospheric air masses around 40°N. Some of these samples identified as influenced by the stratosphere and some as urban (as the air masses below, maybe due to a coarser vertical resolution of the satellite retrieval). CO-rich air masses correspond to those influenced

by biomass burning, mixed-pollution and marine (with CO mixing down near the ocean surface). Whereas these features are clearly depicted by the satellite retrievals, they are not clearly modelled by the reanalyses (Fig. 13c). This is illustrated in Fig. 13c for TCR-2 that show less clear patterns distinguishing the chemical composition air masses. Very similar results are obtained for CAMS reanalysis and MERRA-2 (not shown).


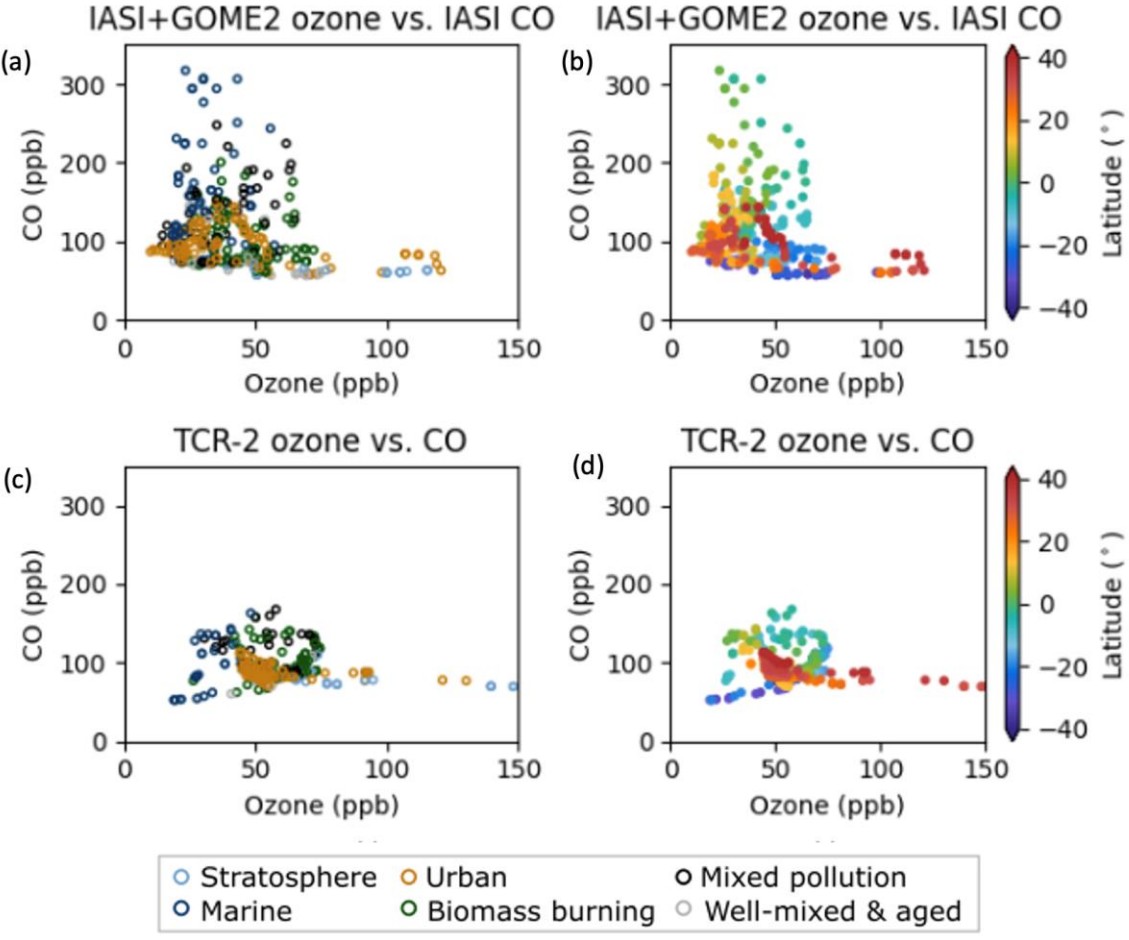



**Figure 13. Scatterplots of CO vs. ozone for air masses sampled by ATom-2 derived from (a and b) satellite measurements of respectively IASI and IASI+GOME2 and (c and d) the TCR-2. Colours indicate (a and c) the air masses classification into six categories (idem as Figs. 11a and 12a) and (b and d) the latitude location.**


The following subsections use the categorization in Fig. 11 to describe the origin of air masses in the Tropical (Sect. 3.2.3.1), Southern (Sect. 3.2.3.2) and Northern (Sect. 3.2.3.3) Atlantic.

### 3.2.3.1 The Tropical Atlantic

As shown in Sect. 3.2.2, large CO enhancement is observed around the equator in four satellite and reanalysis products and
ATom-2 observation (Fig. 9). Chemical tracers suggest that most of these air masses rich in CO are classified into marine air, biomass burning air or mixed pollution air (Fig. 11a). Back trajectory analyses indicate that these CO-enriched air masses in the lower troposphere (15° S–15° N) passed over the Gulf of Guinea, Western, Central and Northern Africa (Figs. 11d). In February, biomass burning is active in Western (south of the Sahara) and Central Africa (Fig. 2a). These results indicate that the biomass burning emissions with rich-CO are transported from the southern part of Western Africa and Central Africa to
the remote Tropical Atlantic via the Gulf of Guinea by southeasterly winds. On the other hand, ozone concentration in the lower troposphere around the equator is moderate according to the four products and ATom-2 observations (Fig. 8). Ozone enhancements are remarked in the Gulf of Guinea and the coastal countries (Fig. 6), while it takes over 6 days for the air masses to travel from the coast to the ATom-2 track far in the Atlantic (Fig. 11d). In this location, the ozone attributed to biomass burning emissions is approximately 7 ppb (~27 %) from the surface to 5 km between 15° S and 15° N, according to
an estimation of the ozone attributed to biomass burning emissions using biomass burning tracer (Fig. 14). This is consistent with significant sinks of ozone often encountered within a lower tropical marine troposphere. In these conditions, the photochemical lifetime of ozone is typically reduced to a few days because of abundant actinic radiation, ample water vapor, and negligible $NO_X$. (e.g., Crutzen et al., 1999). The high abundance of water vapor in the equatorial region is confirmed by ERA5 reanalyses (Fig. 15c).

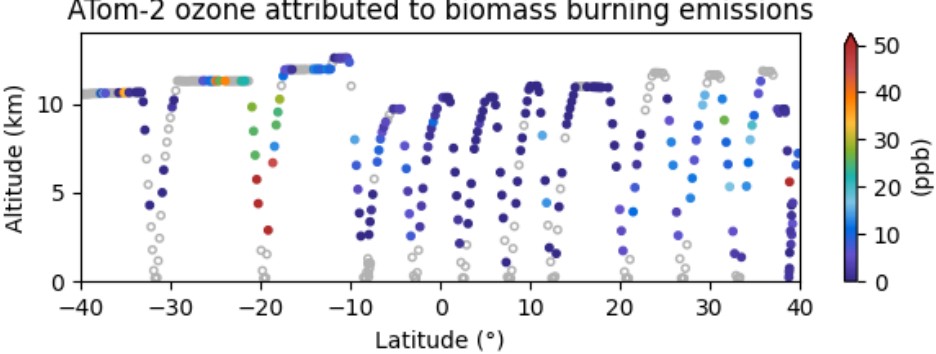




**Figure 14: Vertical profile of ozone concentration attributed to biomass burning.**

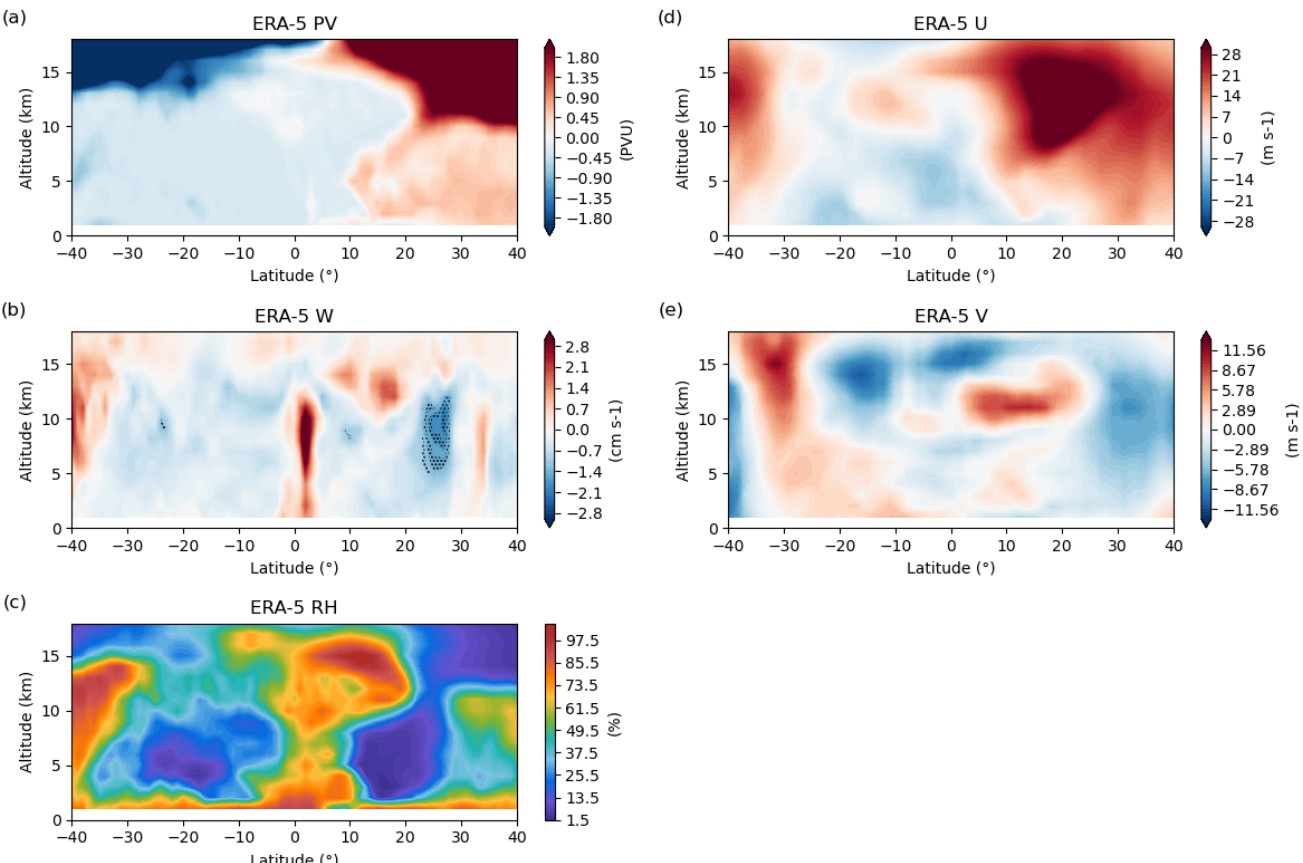

**Figure 15: Vertical profiles of meteorological factors from ERA5 for the period from 13 to 15 February 2017. (a) Potential vorticity,**
**(b) vertical velocity and (c) relative humidity, (d) u and (e) v wind components averaged for the period from 13 to 15 February 2017,**
**and extracted along the flight track. Positive u and v wind components is from the west and the south, respectively.**

Strong upward motion can be observed around the equator (Fig. 15b). High relative humidity can also be observed in the
middle and upper troposphere as well as in the lower troposphere around the equator (Fig. 15c). These results indicate that the
air in the marine boundary layer is uplifted to the upper troposphere by the deep convection, as indicated by collocated OLR
below 220 W m$^{-2}$ (Fig. 1a). Strong downward motion can be observed between 20° N and 30° N (Fig. 15b). It is indicating a
descending branch of a Hadley cell.

Another CO plume collocated with high relative humidity can be identified in the upper troposphere (> 9 km) between 10° S
and 20° N (Fig. 9 and 15c), although not clearly by IASI CO. Most of these CO-rich air masses are classified into biomass
burning air, mixed pollution air or well-mixed and aged air (Fig. 11a). Back trajectory analysis indicates that most of the air





masses are transported from Western and Central Africa to the remote Atlantic via the Gulf of Guinea (Figs. 11e). Some well-mixed and aged air masses show high $NO_X$ concentration and low probability of boundary layer influence (Fig. 11a). This may correspond to the presence of lightning-produced $NO_X$, impacting ozone concentrations. These results suggest that deep convection occurs around the equator, likely injecting water vapor and CO into the upper troposphere from the lower

troposphere. The ozone attributed to biomass burning emissions is approximately 2 ppb (~4%) over 9 km between 10°S and 20°N.

### 3.2.3.2 The Southern Atlantic

At around 20° S in the upper troposphere, the four satellite/reanalyses ozone products and ATom-2 in situ observations clearly depict a stratospheric intrusion (Fig. 8). Very low potential vorticity (< –2 PVU) and downward motion from ERA5 also

indicate downward transport of air masses from the upper troposphere and the lower stratosphere at around 20° S (Fig. 15a–b). The ozone plume extends from the upper troposphere at 20° S to the middle troposphere at the equator according to the reanalyses. Concomitant relative humidity with respect to the ozone plume is low (Fig. 15c), which is typical for stratospheric air. However, this ozone plume is also collocated with a moderate CO plume (Fig. 9), which is not expected for pure stratospheric air. According to the classification shown in Fig. 11a, most of air masses within this plume correspond to

stratosphere air, biomass burning air, well-mixed and aged air, or mixed pollution air. The ozone attributed to biomass burning emissions is approximately 13 ppb (~17 %) over 7 km between 25° S and 5° N and approximately 38 ppb (~50 %) over 3 km between 25° S and 15° S (Fig. 14). Back trajectory analysis indicates that the air is from Central and Eastern Africa (Fig. 11b–c). These results suggest that the ozone plume is influenced by both stratospheric exchanges and biomass burning emissions. Some well-mixed and aged air masses show high $NO_X$ concentration and low probability of boundary layer influence

(Fig. 11a). According to the discussion in Sect. 3.2.3, these conditions suggest the influence of lightning-produced $NO_X$ as ozone precursor. Back trajectories confirm that these lightning-influenced air masses originate from a lightning active region in South America (Fig. 1a).

### 3.2.3.3 The Northern Atlantic

The three reanalyses show a stratospheric intrusion at the UTLS around 25° N, co-located with a descending branch of the Hadley cell, whereas IASI+GOME2 does not show such a feature (Fig. 8) and ATom-2 in a lower clearly lower magnitude (see discussions in Sect. 3.2.1). In this region, we remark high potential vorticity, low relative humidity (Fig. 15a and c) and strong subsidence at altitudes between 6 and 10 km (Fig. 15b). The downward motion speed derived from ERA5 reanalyses is greater than 1 cm s$^{-1}$ in absolute terms, which is significantly larger than the typical value for tropical clear-sky regions

(Gettelman et al., 2004; Das et al., 2016). The CO concentrations collocated with the ozone plume are low (Fig. 10). Although




the origin of these air masses is clearly identified, the magnitude of downward transport of ozone from the stratosphere is likely overestimated by the reanalyses.

Below the UTLS around 20–25° N, back trajectories from the lower and upper troposphere indicate that the air masses originate from North America (Fig. 11f–g). Most of the air mass is classified to urban air in the Northern Hemisphere, especially over

20° N (Fig. 11a). The influence of biomass burning is seen closer to the equator. These results suggest that the reanalyses likely overestimate the urban influence of the Northern Hemisphere over the Atlantic.

## 4 Conclusions

We have presented an analysis of the tropospheric ozone spatial distribution and its origins using satellite (IASI+GOME2), in situ observations (ATom-2) and three global reanalyses (TCR-2, CAMS reanalysis and MERRA-2) over the Tropical and

South Atlantic in February 2017. Seasonal variation of regional discrepancies (expressed as spatial standard deviations) between the satellite observations and the reanalyses of monthly ozone distribution over the Tropical Atlantic (25° S–25° N, 34° W–18° W) show a clear seasonality corresponding to two biomass burning seasons and two transition seasons in Africa. The largest differences between these datasets are seen for the months of January and February. In this last period, the region is likely influenced by biomass burning emissions form the southern part of West Africa and Central Africa and deep

convection over the Gulf of Guinea (over the Ocean), Central and Southern Africa (as depicted by the frequent lightning activity). Comparisons of horizontal distributions of monthly ozone in the lowermost troposphere (surface–3 km) in February 2017 show that IASI+GOME2, TCR-2 and CAMS reanalysis datasets display high concentration of ozone from Western Africa to the Gulf of Guinea. MERRA-2 shows no enhancement of ozone in Western Africa probably because of the use of a simplified chemistry. Only IASI+GOME2 depicts an enhancement of lowermost tropospheric ozone north of the St. Helena

anticyclone. In the middle and upper troposphere, all products show a horseshoe-shaped structure of high concentration of ozone from Southern and Western Africa to the east of Brazil. IASI+GOME2 show lower ozone concentration in the South America compared to three chemistry reanalysis products.

The tropospheric ozone spatial distributions are generally similar for the monthly mean in February 2017 and the average in the 3-day period when ATom-2 in situ measurements are available (13–15 February 2017). We analyse these vertical profiles

along a south-north track from 40° S to 40° N to assess the capability of satellite and chemistry reanalysis products to characterize the spatial distribution of the tropospheric ozone over the Tropical Atlantic. We clearly observe that only the IASI+GOME2 satellite approach is able to describe the strong gradient of significant tropospheric ozone enhancements in the Southern Hemisphere and low abundances in the Northern Hemisphere, in agreement with ATom-2 in situ measurements. An enhancement of ozone in the north of the St. Helena anticyclone between 10° S and 20° S is consistently observed by both

IASI+GOME2 and ATom-2, whereas the three reanalyses show low ozone values below 20 ppb. North of the equator, all three reanalyses particularly fail to depict the weak ozone concentrations consistently observed by satellite and in situ sensors



(IASI+GOME2 and ATom-2 agree both in absolute values and correlation). This is partly explained by a significant ozone enhancement displayed by three chemistry reanalyses in the descending branch of the Hadley cell at around 25° N, which is not depicted by IASI+GOME2 nor ATom-2. Back trajectory analysis indicates that these air masses in the lower and upper

troposphere originate from North America.

Using ozone and CO abundances from satellite and in situ measurements, together with measurements of several tracers, we describe the sources of ozone and CO plumes over the tropical Atlantic (see the scheme in Fig. 16). Around the equator, an CO plume with low-ozone is observed in the lowermost troposphere by all datasets (satellite observations, in situ measurements and three reanalyses). Most of air masses is classified into marine air (near the ocean), biomass burning air or mixed pollution

air and passed over the Gulf of Guinea, Western, Central and Northern Africa. Far from the coast (10° S, 15° W) in the ATom-2 flight track, the amount of ozone attributed to biomass burning emissions is low with the value of approximately 7 ppb (~27 %) from the surface to 5 km between 15° S and 15° N. This is likely link to active sinks of tropospheric ozone during the transport from the continent due to the high relative humidity.

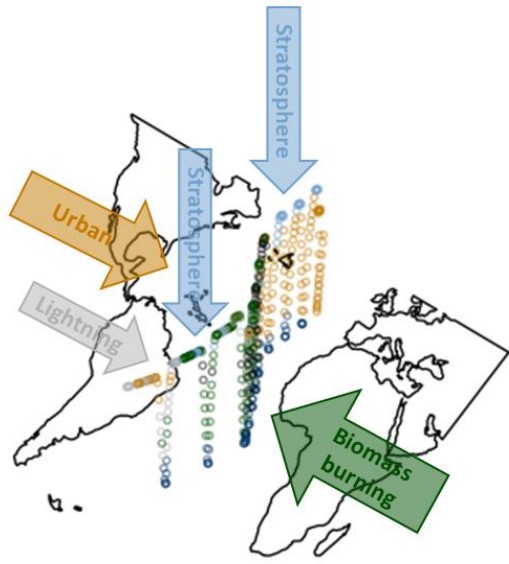

**Figure 16: Schematic of mechanisms that ozone and CO concentrations over the South and Tropical Atlantic for the period from 13 to 15 February 2017.**

Strong upward motion around the equator injects water vapor and CO into the upper troposphere from the lower troposphere. Most of air masses within the CO plume in the upper troposphere (> 9 km) between 10° S and 20° N is classified as influenced

by biomass burning, mixed pollution air or well-mixed and aged air. This CO plume is transported from Western and Central Africa to the remote Atlantic via the Gulf of Guinea. In addition, some well-mixed and aged air masses might be influenced





by lightning-produced $NO_X$ indicated by high $NO_X$ concentration and low probability of boundary layer influence. The ozone attributed to biomass burning emissions is approximately 2 ppb (~4 %).

At around 20 °S in the upper troposphere, all datasets show a stratospheric intrusion. An ozone plume with low-relative humidity and high-CO transported from Central and Eastern Africa is observed in the middle and upper troposphere. In this region, the ozone plume in the lower troposphere only depicted IASI+GOME2 and ATom-2 north of St. Helena anticyclone is classified as stratosphere air, biomass burning air, well-mixed and aged air, or mixed pollution air. The ozone attributed to biomass burning emissions is approximately 13 ppb (~17 %) over 7 km between 25° S and 5° N and approximately 38 ppb (~50 %) over 3 km between 25° S and 15° S. In addition, lightning-produced $NO_X$ might also impact on the ozone 655 concentration.

Air masses in the Northern Hemisphere with weak concentrations of ozone (over 20° N) are classified as influenced by urban sources (from North America according to back trajectories). These results suggest that the reanalyses overestimate the abundance of tropospheric ozone over the remote locations in the Tropical Atlantic for air masses influenced by urban sources of North America in February 2017.


### *Data availability*

The IASI+GOME2 ozone and the IASI CO datasets derived from MetOp-B global measurements are available on the French data centre AERIS (https://iasi.aeris-data.fr/).

The TCR-2 dataset is available at https://tes.jpl.nasa.gov/tes/chemical-reanalysis/

The CAMS reanalysis dataset has been downloaded at the Atmosphere Data Store (ADS) (https://ads.atmosphere.copernicus.eu/).

MERRA-2 data are distributed by the NASA Goddard Earth Sciences (GES) Data and Information Services Center (DISC) (https://disc.gsfc.nasa.gov/).

The ATom data are distributed by the Oak Ridge National Laboratory Distributed Active Archive Center (ORNL DAAC) 670 (https://daac.ornl.gov/).

ERA5 data have been downloaded from the Climate Data Store (https://cds.climate.copernicus.eu/).

The WWLLN Global Lightning Climatology (WGLC) global gridded lighting timeseries is available at https://github.com/ARVE-Research/WGLC.

The monthly OLR data is distributed by the National Oceanic and Atmospheric Administration (NOAA) Physical Science 675 Laboratory (PSL) (https://psl.noaa.gov/).

The active fire products (MCD14ML Collection 6) is distributed by the Fire Information for Resources Management System (FIRMS) (https://firms.modaps.eosdis.nasa.gov/).

### *Author contributions*



SO and JC conducted the research work and lead the writing of the main manuscript. JC provided the IASI+GOME2 satellite
       data. CB provided support in the production of IASI+GOME2 data. KM provided the TCR-2 tropospheric chemistry reanalysis
       data. All authors (SO, JC…) contributed to the discussions, refinement of the results and improvements of the paper.

*Competing interests*

The contact author has declared that none of the authors has any competing interests.

*Acknowledgements*

Authors acknowledge the financial support of the Centre National des Etudes Spatiales (CNES, the French Space Agency) via
the SURVEYPOLLUTION and TOTICE research projects from the TOSCA (Terre Ocean Surface Continentale Atmosphère)
committee, the Université Paris Est Créteil (UPEC), and the Centre National des Recherches Scientifiques–Institut National
des Sciences de l'Univers (CNRS-INSU), for helping in achieving this research work and its publication. We also acknowledge
the AERIS data centre for providing IASI+GOME2 ozone and IASI CO (developed by the Université Libre de Bruxelles and
the LATMOS laboratory) datasets, the ADS for providing CAMS reanalysis dataset, the GES DISC providing the MERRA-2
datasets, the CDS for providing ERA5 datasets, the ORNL DAAC for providing the ATom datasets, the WWLLN providing
the WGLC lightning timeseries, NOAA/PSL for providing the OLR data, and the FIRMS for providing the active fire products.
We also acknowledge the support of the NASA Atmospheric Composition: Aura Science Team Program (19-AURAST19-
0044), Earth Science U.S. Participating Investigator program (22-EUSPI22-0005), ACMAP (22-ACMAP22-0013), and the
NASA TROPESS project. Part of this work was conducted at the Jet Propulsion Laboratory, California Institute of Technology,
under contract with NASA. Ilann Bourgeois is also acknowledged for providing the method of tracers.

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
