# Peer review of "Natural and anthropogenic influence on tropospheric ozone variability over the Tropical Atlantic unveiled by satellite and in situ observations"

_EGUsphere, 2024_

## Community Comment (CC1)

February 4, 2025

Comments by Owen R. Cooper (TOAR Scientific Coordinator of the Community Special Issue) on:

**Natural and anthropogenic influence on tropospheric ozone variability over the Tropical Atlantic unveiled by satellite and in situ observations**

Sachiko Okamoto, Juan Cuesta, Gaëlle Dufour, Maxmim Eremenko, Kazuyuki Miyazaki, Cathy Boonne, Hiroshi Tanimoto, Jeff Peischl, and Chelsea Thompson

EGUsphere [preprint], https://doi.org/10.5194/egusphere-2024-3758
Discussion started: 20 Dec 2024
Discussion closes:   7 Feb 2025

This review is by Owen Cooper, TOAR Scientific Coordinator of the TOAR-II Community Special Issue. I, or a member of the TOAR-II Steering Committee, will post comments on all papers submitted to the TOAR-II Community Special Issue, which is an inter-journal special issue accommodating submissions to six Copernicus journals:  ACP (lead journal), AMT, GMD, ESSD, ASCMO and BG. The primary purpose of these reviews is to identify any discrepancies across the TOAR-II submissions, and to allow the author teams time to address the discrepancies.  Additional comments may be included with the reviews. While O. Cooper and members of the TOAR Steering Committee may post open comments on papers submitted to the TOAR-II Community Special Issue, they are not involved with the decision to accept or reject a paper for publication, which is entirely handled by the journal's editorial team.

**Comments regarding TOAR-II guidelines:**

TOAR-II has produced two guidance documents to help authors develop their manuscripts so that results can be consistently compared across the wide range of studies that will be written for the TOAR-II Community Special Issue.  Both guidance documents can be found on the TOAR-II webpage:
https://igacproject.org/activities/TOAR/TOAR-II

*The TOAR-II Community Special Issue Guidelines*:   In the spirit of collaboration and to allow TOAR-II findings to be directly comparable across publications, the TOAR-II Steering Committee has issued this set of guidelines regarding style, units, plotting scales, regional and tropospheric column comparisons, and tropopause definitions.

*The TOAR-II Recommendations for Statistical Analyses*:  The aim of this guidance note is to provide recommendations on best statistical practices and to ensure consistent communication of statistical analysis and associated uncertainty across TOAR publications. The scope includes approaches for reporting trends, a discussion of strengths and weaknesses of commonly used techniques, and calibrated language for the communication of uncertainty. Table 3 of the TOAR-II statistical guidelines provides calibrated language for describing trends and uncertainty, similar to the approach of IPCC, which allows trends to be discussed without having to use the problematic expression, "statistically significant".

**General comments:**

This analysis evaluates three chemical reanalysis products above the tropical North and South Atlantic Oceans, with a focus on February 2017. Some discussion needs to be provided to state why the analysis only focuses on February 2017, when plenty of in situ observations are available for other seasons and years. For example, why not provide additional analysis of in situ observations for the months of September-October-November? This is the time of year when the well-known ozone maximum occurs above the South Atlantic (Thompson et al., 2021).

A paper recently published in the TOAR-II Community Special Issue (Gaudel et al., 2024) provides an extensive evaluation of several satellite products across the tropics. This paper takes advantage of 25 years of in situ observations, including several ATOM flights, 8 ozonesonde sites, and thousands of IAGOS aircraft profiles above five regions. These same data sets can be applied to your study. In particular the NASA SHADOZ ozonesonde station on Ascension Island reveals the seasonal ozone variability in the center of the South Atlantic ozone maximum (Thompson et al., 2021). This station also has several years (2016-2019) of surface ozone observations. The SHADOZ data archive is here: https://tropo.gsfc.nasa.gov/shadoz/Archive.html

Finally, ozone production and transport above the tropical Atlantic has been studied for decades (Fishman et al., 1986, 1991; Thompson et al., 1996, 2000; Moxim and Levy, 2000), and it is important for the authors to describe their findings in relation to previous work, and to clearly state what is new about their findings.

**Specific Comments:**

Lines 640-641
I don't understand the caption to Figure 16 as the sentence seems to be missing a verb.

Fishman et al. 1983 appears in the list of references, but I don't see this paper cited in the main text

**References**

Fishman, J., Minnis, P. and Reichle Jr, H.G., 1986. Use of satellite data to study tropospheric ozone in the tropics. Journal of Geophysical Research: Atmospheres, 91(D13), pp.14451-14465.

Fishman, J., Fakhruzzaman, F., Cros, B., & Nganga, D. (1991). Identification of widespread pollution in the southern hemisphere from satellite analyses. Science, 252(5013), 1693–1696. https://doi.org/10.1126/science.252.5013.1693

Gaudel, A., Bourgeois, I., Li, M., Chang, K.-L., Ziemke, J., Sauvage, B., Stauffer, R. M., Thompson, A. M., Kollonige, D. E., Smith, N., Hubert, D., Keppens, A., Cuesta, J., Heue, K.-P., Veefkind, P., Aikin, K., Peischl, J., Thompson, C. R., Ryerson, T. B., Frost, G. J., McDonald, B. C., and Cooper, O. R. (2024), Tropical tropospheric ozone distribution and trends from in situ and satellite data, Atmos. Chem. Phys., 24, 9975–10000, https://doi.org/10.5194/acp-24-9975-2024

Moxim, W. J., & Levy, H. (2000). A model analysis of the tropical South Atlantic Ocean tropospheric ozone maximum: The interaction of transport and chemistry. Journal of Geophysical Research, 105(D13), 17393–17415. https://doi.org/10.1029/2000jd900175

Thompson, A. M., Pickering, K. E., McNamara, D. P., Schoeberl, M. R., Hudson, R. D., Kim, J. H., et al. (1996). Where did tropospheric ozone over southern Africa and the tropical Atlantic come from in October 1992? Insights from TOMS, GTE TRACE A, and SAFARI 1992. *Journal of Geophysical Research*, 101(D19), 24251–24278. https://doi.org/10.1029/96jd01463

Thompson, A.M., Doddridge, B.G., Witte, J.C., Hudson, R.D., Luke, W.T., Johnson, J.E., Johnson, B.J., Oltmans, S.J. and Weller, R., 2000. A tropical Atlantic paradox: Shipboard and satellite views of a tropospheric ozone maximum and wave-one in January–February 1999. Geophysical Research Letters, 27(20), pp.3317-3320.

Thompson, A. M., Stauffer, R. M., Wargan, K., Witte, J. C., Kollonige, D. E., & Ziemke, J. R. (2021). Regional and seasonal trends in tropical ozone from SHADOZ profiles: Reference for models and satellite products. Journal of Geophysical Research: Atmospheres, 126, e2021JD034691. https://doi.org/10.1029/2021JD034691

---

## Community Comment (CC2)

Comments on Okamoto et al (2024)

This paper presents a study characterizing the vertical and horizontal distribution of tropospheric ozone over the South and Tropical Atlantic during February 2017 using a combined UV-IR product from IASI+GOME2, in situ airborne measurements from the Atmospheric Tomography Mission (ATom) and 3 reanalysis models.

The topic is important, but there are a few items that should be addressed as described below.

1) The observational data used within this study is very limited and should be expanded to include other vertical profiles available for your period of interest. Thompson et al (2021), which should be cited in this paper (and more recently Thompson et al (2025), presents a great reference for models and satellite products in the tropics, including the Atlantic, based on multiple SHADOZ ozonesonde stations within your region of interest. These stations include Ascension Island, UK, and Natal, Brazil, where vertical ozone profiles from 1998-2023 are available online here: https://tropo.gsfc.nasa.gov/shadoz/Archive.html. Also available during February 2017 is in situ ozone monitoring data from the Ascension Island station: https://tropo.gsfc.nasa.gov/shadoz/Ascension.html. It is important to provide insight from ground-based measurements for your tropospheric ozone discussion outside the limited aircraft flights and one satellite product.

2) Recent studies using ground-based and satellite observations discussing current tropical tropospheric ozone distributions should be addressed and cited: Thompson et al (2021; 2025) and Gaudel et al. (2024).

3) Based on the knowledge that UV/IR satellite products are limited with their sensitivity in the lower troposphere (using 0-3km layer here) and over water (versus land), have you done comparisons with similar tropospheric ozone satellite products? A demonstration of what the IASI-GOME2 vertical profiles, and other similar satellite products, look like over a tropical Atlantic site like Ascension Island for your period of interest would show the full extent of the vertical ozone distribution and any limitations in the presented satellite and model reanalysis. Pennington et al (2024) describes a joint AIRS/OMI ozone product similar to the IASI-GOME2 that could be used and Keppens et al (2025) provides a list of current tropospheric ozone observing satellite products that could be compared here.

Gaudel, A., Bourgeois, I., Li, M., Chang, K.-L., Ziemke, J., Sauvage, B., Stauffer, R. M., Thompson, A. M., Kollonige, D. E., Smith, N., Hubert, D., Keppens, A., Cuesta, J., Heue, K.-P., Veefkind, P., Aikin, K., Peischl, J., Thompson, C. R., Ryerson, T. B., Frost, G. J., McDonald, B. C., and Cooper, O. R.: Tropical tropospheric ozone distribution and trends from in situ and satellite data, Atmos. Chem. Phys., 24, 9975–10000, https://doi.org/10.5194/acp-24-9975-2024, 2024.

Keppens, A., Hubert, D., Granville, J., Nath, O., Lambert, J.-C., Wespes, C., Coheur, P.-F., Clerbaux, C., Boynard, A., Siddans, R., Latter, B., Kerridge, B., Di Pede, S., Veefkind, P., Cuesta, J., Dufour, G., Heue, K.-P., Coldewey-Egbers, M., Loyola, D., Orfanoz-Cheuquelaf, A., Maratt Satheesan, S., Eichmann, K.-U., Rozanov, A., Sofieva, V. F., Ziemke, J. R., Inness, A., Van Malderen, R., and Hoffmann, L.: Harmonisation of sixteen tropospheric ozone satellite data records, EGUsphere [preprint], https://doi.org/10.5194/egusphere-2024-3746, 2025.

Pennington, E. A., Osterman, G. B., Payne, V. H., Miyazaki, K., Bowman, K. W., and Neu, J. L.: Quantifying biases in TROPESS AIRS, CrIS, and joint AIRS+OMI tropospheric ozone products using ozonesondes, EGUsphere [preprint], https://doi.org/10.5194/egusphere-2024-3701, 2024.

Thompson, A. M., Stauffer, R. M., Wargan, K., Witte, J. C., Kollonige, D. E., & Ziemke, J. R., Regional and seasonal trends in tropical ozone from SHADOZ profiles: Reference for models and satellite products (2021), Journal of Geophysical Research: Atmospheres, 126, http://doi.org/10.1029/2021JD034691.

Thompson, A. M., Stauffer, R. M., Kollonige, D. E., Ziemke, J. R., Cazorla, M., Wolff, P., and Sauvage, B.: Tropical Ozone Trends (1998 to 2023): A Synthesis from SHADOZ, IAGOS and OMI/MLS Observations, EGUsphere [preprint], https://doi.org/10.5194/egusphere-2024-3761, 2025.

---

## Author Comment (AC1)

We are grateful to the editor and reviewers for dedicating their valuable time and effort to reviewing our manuscript. The comments raised by the reviewers have significantly improved the quality and clarity of our work. We agree with most of the provided feedback and accordingly we have made several changes to the manuscript. For clarity, we first outline some of the major changes made before proceeding with a point-by-point response to the reviewers' comments. We would have liked to include as much as possible additional datasets suggested by the anonymous referee #1 (RC1) and the community comments. However, for sake of briefness and clarity as very clearly pointed by the anonymous referee #1 (RC1), we have made the choice to add substantial additional datasets but could not add every dataset. The choice was made to keep the consistency of the paper and enlarge the time coverage of the analyses, while focusing on the main objective of the paper linked to a description of tropospheric ozone and its origin over the Tropical Atlantic. Please note that the reviewers' comments are in black italic, while our responses are in blue. Any modified or additional text in the revised manuscript is highlighted in red.

1)      Excluding MERRA-2

We have excluded MERRA-2 from our intercomparison study as suggested by Dr. Kenneth Pickering (CC3). This is indeed consistent with the fact that the MERRA-2 ozone product is recommended for the upper troposphere and stratosphere, while we focus on the distribution of tropospheric ozone and down to its lower part.

2)      Adding another case study during 17–20 October 2017 (ATom-3)

As pointed out by most of reviewers, our study was based on the analysis during 13–15 February 2017 (corresponding to the ATom-2 field campaign). We agree with reviewers that the comparison with additional in situ data could help to draw more general conclusions on the performance of these products. In that sense, we have added a whole new study case for investigating horizontal and vertical ozone distributions during 17–20 October 2017 (during the ATom-3 field campaign). The detailed characterization of two contrasting season allows a very interesting description of two periods with intense fires either in the Northen Hemisphere (February) over West Africa or in the Southern Hemisphere (October) over Central Africa. As shown in Fig. S4, ozone behavior is also different with the highest average concentrations and a moderate coefficient of variation (CV) in October and very large CV in February. These two seasons also show a contrast with respect to the ITCZ location and the major convective activity. The fact of choosing two case studies is a good compromise allowing a detailed description of each of them (similar aim as in the first submission of the paper, which considered one case) while keeping the manuscript of a reasonable length (as requested by the anonymous referee #1). We think that the contrast between ATom-2 and ATom-3 periods emphasizes our original conclusions. Consequently, Section 3 of the manuscript has been substantially revised and the updated results have been reflected in the Conclusion section of the revised manuscript.

3)      SHADOZ ozonesonde profiles

We decide not to include the SHADOZ ozonesonde profiles for the analysis. We totally agree that SHADOZ ozonesonde profiles have provided useful information to unveil distributions of ozone over the Atlantic. However, there are insufficient observations at three SHADOZ stations in February 2017, which we do not consider as sufficient to draw significant conclusions (particularly accounting with possible differences in co-location and differences in scale). We find the Atom-2 and Atom-3 in situ airborne sounding of ozone and other relevant variables with a large spatial scale as key information comparable with IASI+GOME2 satellite measurements, which is extensively used in the manuscript.

|  | Number of data in February 2017 | Number of data in October 2017 |
| --- | --- | --- |
| Ascension Is., UK | 0 | 5 |
| Natal, Brazil | 1 | 2 |
| Paramaribo, Surinam | 3 | 4 |

**Author Response to RC1 (Anonymous Referee #1)**

*In "Natural and anthropogenic influence on tropospheric ozone variability over the Tropical Atlantic unveiled by satellite and in situ observations" Okamoto et al. discuss ozone levels over the Atlantic Ocean based on satellite observations, reanalysis products and in situ observations from the ATom 2 campaign in February 2017. The authors show that ozone can be attributed to various different sources in the studied region including biomass burning, urban influences and stratospheric intrusion. They report an overprediction of ozone in lower altitudes in all investigated reanalysis products.*

*This is an interesting study. However, I have several questions and comments, which need to be addressed before I can recommend this paper for publication. The manuscript is very long and contains too many figures. I recommend creating a Supplement and moving some figures and explanations there. Overall, the manuscript would benefit from being more concise. I further suggest including the other three ATom campaigns in this study. The authors investigate seasonality and as the ATom deployments took place in all seasons and all cover the studied region, the in situ data will be a valuable addition. While the authors present the differences in the reanalysis products for only two days in February 2017, the comparison with further in situ data could help to draw general conclusions on the performance of these products. Please find my detailed comments in the following.*

**Response:** Agreed and added. We are very grateful to the reviewer for thoroughly reviewing our manuscript and for providing valuable insights to improve it. As suggested, we have chosen to add an additional contrasting case study with new in situ observations (those of ATom-2), while keeping the manuscript reasonably concise (as clearly requested Anonymous Referee #1) by moving several figures to the supplementary material. The new case study based on ATom-3 corresponds to 17–20 October 2017. The analysis of two contrasting periods/seasons is very rich, by being opposite in the year, February and October, and in atmospheric conditions and sources of ozone precursors. They cover the two contrasting situations in Africa with fires in either the Northern or Southern Hemisphere. The study of these two periods which are key for the tropospheric ozone distribution in the region emphases our conclusions. At the same time, this choice allows keeping the clearness of the paper and its detailed analyses, which remaining reasonable the length and size of the figures (currently 13 figures with already several panels). The study of other periods or cases will be addressed in future work. Please see page 1 of this document. The relevant texts and figures have been added or modified in Section 3. Some figures have been moved to the supplement.

*Lines 35 f.: Stratospheric intrusion only contributes a small fraction to the overall tropospheric ozone (Lelieveld & Dentener, 2000, doi: 10.1029/1999JD901011)*

**Response:** Agreed and clarified. We have added the description about quantitative contribution from stratospheric intrusion, representing approximately 1/10 of that chemically produced in the troposphere (lines 37–38 on page 2).

The net influx ozone from the stratosphere was estimated at $552 \pm 168$ Tg, and is smaller than the amount of chemical production ($5110 \pm 606$ Tg; Young et al., 2013).

*Line 41 f.: biogenic sources? – There seems to be a word missing. I further recommend briefly discussing the formation mechanism of ozone from these precursors here (or in the previous paragraph).*

**Response:** Agreed and added. It is indeed biogenic sources (added in the manuscript). We have also extended the description about ozone precursor sources (lines 33–36 on page 2).

These ozone precursors originate from both anthropogenic (fossil fuel combustion in power plants, industrial activities, transportation and crop burning) and natural sources (wetland $CH_4$ emissions, wildfires, biogenic hydrocarbon emissions, lightning and soil $NO_X$ emissions) (e.g., Elshorbany et al., 2024).

*Lines 45: MOZAIC?*

**Response:** Yes. We have corrected this typo.

*Line 48: Several studies have shown that lightning is the dominant source of ozone in the upper troposphere, e.g. Schumann & Huntrieser, 2007 (doi: 10.5194/acp-7-3823-2007) Nussbaumer et al., 2023 (doi:10.5194/acp-23-12651-2023). Please consider citing this literature.*

**Response:** Agreed and cited.

*Line 54: Over the tropics, 9-13km is still tropospheric. The UTLS region only applies to Southern or Northern Extratropical Latitudes.*

**Response:** Agreed and rephrased (line 80–82 on page 3).

However, most of IAGOS data is acquired in the extratropical upper troposphere/lower stratosphere (UTLS) and in the tropical upper troposphere when the aircraft attain cruising altitude in the altitude band of 9–13 km (See S1 in the Supplement).

Line 57: What do the authors mean by "the observational gap of air pollution" and how does it relate to the tropical Atlantic?

**Response:** Agreed and supressed (line 84 on page 3). As air quality is not an issue over the tropical Atlantic, we do not mention air pollution in the sentence of the revised manuscript.

Satellite observations offer a great potential to overcome the limited spatial coverage of ground-based measurements.

*Line 69: What "major outbreaks" are the authors referring to?*

**Response:** Agreed and rephrased (line 97 on page 4). We refer to transboundary pollution events across east Asia in the revised manuscript.

Air-quality-relevant capabilities of IASI+GOME2 have been demonstrated by quantitatively describing the transport pathways, the daily evolution, and photochemical production in the lowermost troposphere during transboundary ozone pollution events across east Asia (Cuesta et al., 2018) and Europe (Cuesta et al., 2013; 2022; Okamoto et al., 2023).

*Lines 181 f.: Please briefly describe the measurements / instruments including uncertainties and detection limits of the trace gases used in this analysis.*

**Response:** Clarified. The measurement methods of the tracers are extensively described in Bourgeois et al. (2021). Therefore, we have decided not to describe the details in this paper (lines 206–207 on page 8).

The measurement methods of these tracers are described in detail elsewhere (Bourgeois et al., 2021).

*Lines 187 f.: "(…) they use a pair of HCN biomass of a burning tracer and $C_2Cl_4$ of an urban tracer." This sentence is difficult to read, I recommend rephrasing.*

**Response:** Rephrased (lines 211–212 on page 8). There were missing parenthesis.

To quantify the respective influence of biomass burning and urban emissions on each air parcel, they use a pair of HCN (biomass burning tracer) and $C_2Cl_4$ (urban tracer).

*Line 190: The conditions for polluted and aged air are the same. I assume the first "<" should be ">" instead? Is well-mixed and aged air equivalent to clean / unpolluted air? What's the lifetime of HCN and C2Cl4 in the troposphere?*

**Response:** Corrected. The inequality sign of the definition for well-mixed and aged air were corrected. In the first case, the sign was correction to ">". Indeed, what we mention as "well-mixed and aged air" corresponds to rather clear or background conditions as both tracers are lower the regional values. This is indicate in line 214–218 on page 8).

For the region (40° S–40° N), air parcels are either defined as urban air (urban tracer > regional median, biomass burning tracer < regional median), biomass burning air (biomass burning tracer > regional median, urban tracer < regional median), mixed pollution air (both urban and biomass burning tracers > regional median), and well-mixed and aged air (both urban and biomass burning tracers < regional median) corresponding to rather clean or background conditions.

Mean global lifetime of HCN is 150 days and that of $C_2Cl_4$ is 101 days, thus pretty long-range transport tracers and consistent with CO. This information is newly indicated in the paragraph (see next comment).

Line 195: Are the lifetimes of X and CO similar? If not, does this introduce errors in this method due to different transport ranges?

**Response:** Bourgeois et al. (2021) chose HCN and $C_2Cl_4$ because their lifetime is similar to that of CO. The mean global lifetime is 3 months (CO), 3 and half months ($C_2Cl_4$) and 5 months (HCN). This information is provided in the revised manuscript (line 212–214 on page 8).

These two tracers are chosen because their lifetime is similar to that of CO (being between three and five months for these three tracers), and they have been used as a tracer in the previous literature.

*Line 198: I recommend adding the definition of the emission ratio here as well.*

**Response:** Added (line 226 on page 9).

Bourgeois et al. (2021) use 5.7 pptv ppbv$^{-1}$ (HCN/CO) and 0.03 pptv ppbv$^{-1}$ ($C_2Cl_4$/CO) as ERs of biomass burning and urban air according to Andreae (2019) and Kondo et al. (2004).

*Line 202: Is this equation used for all measurements and if yes, why were the air masses classified into the four categories before? Or is it only used to distinguish the sources of ozone in air masses defined as "mixed pollution air"?*

**Response:** Clarified. This equation is used for measurements corresponding to airmass identified as originating from urban or biomass burning conditions as compared to background conditions (mixed and aged air masses). These are the deltas in the equation. According to this, we clearly need the identification of air masses into the four categories before being able to calculate NEMRx. Results are presented in Fig.13.

*Figure 2: This manuscript has many Figures. I recommend choosing one or two months (for example February and August) and showing the lightning intensity and the fire radiation power in one Figure. The remaining panels can be shown in a Supplement. This makes the comparison of the location of maximum lightning and maximum fire activity easier and reduces the number of Figures.*

**Response:** Agreed and done. We have chosen two months (February and October which are coincident with ATom-2 and ATom-3 in the rest of the manuscript) for condensing previous Figures 1-2 into the new Fig. 1. Detailed distributions have been moved to the supplement (Figs. S2 and 3).

*Line 291: Do the coordinates describe a box and therefore also include continental regions? If yes, does it maybe make more sense to only consider the maritime regions? It could help to add an outline of the discussed area in the Supplement or as a subpanel in Figure 3.*

**Response:** Clarified. We have added a black rectangle in Figs. 2 and 3. This region does not contain continental areas, but only maritime regions.

*Figure 3: Is the SD by itself really meaningful in regard to the uncertainty of the reanalysis products / satellite observations? Maybe it would make more sense to look at the SD relative to the O$_3$ measurement (SD/O$_3$value*100%). I also recommend adding the mean monthly O$_3$ in the Figure.*

**Response:** Thank you for the suggestion. Done. We have replaced plots of SD by those of CV in relative terms in % (Figs. 2, 3 and S4) and have added monthly mean ozone in Fig. S4.

*Line 332: How was this altitude range chosen? The chemical composition at 6 km vs 12km is very different. While upper tropospheric ozone is strongly impacted by deep convective updraft as well as lightning activity, the free troposphere is rather decoupled from the surface and is not impacted by lightning. I recommend looking at the free troposphere, e.g. 3 - 10km, and the upper troposphere, e.g. >10 km, separately.*

**Response:** Thank you for your comment. Clarified and done. In order to analyse ozone distribution in the layer above 6 km but being less influenced by deep convection and lighting at the UTLS (Upper Troposphere Lower Stratosphere), we have changed the altitude range in Figs. 3 and 5 to 6-9 km (instead of 6–12 km).

*Section 3.2: This section makes me wonder why the authors limit themselves to one of the ATom deployments. The four deployments cover the same area during all seasons and could provide some more insights into the discrepancies with the reanalysis products.*

**Response:** Agreed and done. We have added a whole new analysis of ATom-3 datasets in October 2017. For avoiding enlarging the paper and number of figures too much, we concentrated in ATom-2 and ATom-3 with contrasting interesting conditions. Please see page 1 of this document for further details on this choice.

*Figure 6 + 7: These Figures (including their description in the text) are a repetition of Figures 4 and 5 and I recommend deleting them or moving them to the Supplement.*

**Response:** Clarified. The aim of these figures (currently Figs. 2 and 3) is to clearly show that the detailed study of figures 4 to 13 does not only correspond specific situations for the 3 days of the campaigns (ATom-2/3) but describes a prevailing situation in terms of tropospheric ozone regional distribution during the whole months of February and October 2017. This is an important message that needs to be in the body of the manuscript. Therefore, we have decided not to move these figures to Supplement, showing the clear transition in the study from monthly/seasonal conditions to the situations observed in the field campaigns ATom-2/3.

*Lines 400 ff.: It is difficult to see the differences and similarities between the in situ observations and the satellite / reanalysis products just by the color. Maybe instead (or additionally) a vertical profile with averaged ozone could be used (Altitude as y-axis and Ozone mixing ratio as x-axis). Or an additional column could be added to the Figure showing the difference between the ATom2 data and the other products in each panel.*

**Response:** Clarified. It is important to note that these transects cover a very large region extending from south to north over 80 degrees of latitude, thus more than 8000 km with a large variety of conditions from the two hemispheres, affecting these airmass. Withdrawn the latitudinal dimension would withdraw the geographical information and would reduce the information to an evaluation of resemblance between two datasets. Therefore, we have decided to keep the plot style to that of transects. The style of figures using the same colorscale for both datasets allow to distinguish between discrepancies (sharp changes in colors) to similarities (something both datasets show very similar values), which is indeed the message we would like to provide. If we change the size of style of the marker for the in situ data, would leave less space to the background transect and reduce the clarity of the figure (we have tested several sizes so as to have the good compromise of visibility of each dataset).

*Line 407: "in the" is repeated.*

**Response:** Corrected.

*Table 2: Why was only the latitude band between 10°N and 30°N investigated?*

**Response:** Clarified. In this specific table, we have selected this latitude band in the Northern Hemisphere (10° N–30° N) to analyse the specific discrepancy between reanalyses and in situ measurements, for a region in good agreement with the satellite observations. These last ones also show a much broader regional picture of the differences, thus a key information from them.

*Figure 9: Similar to O₃, it is difficult to rank the agreement between the ATom2 data and the other products just by color. I recommend using a more quantitative method.*

**Response:** Clarified in comment the response to the comment starting in "Lines 400 ff." (3 comments above).

Line 466: "not depicted by any"?

**Response:** Corrected (line 533 on page 27).

On the other hand, the two reanalyses display a second maximum around 20° N (Fig. 8d and e), which is not depicted by any of the observational datasets (satellite or in situ).

*Figure 10: See comments above.*

**Response:** Idem comment "Figure 9…" (2 comments above).

Lines 482: In the upper troposphere, lightning NOx is the most important source of O₃. Could it make more sense to define lightning as an individual category?

**Response:** Clarified. This distinction between NOx originating from lightning and that emitted by anthropogenic sources in the boundary layer is already done in the manuscript. We do this specific analysis in the paragraphs (line 580–585 on page 31) and in Fig. 11. The way we do such discrimination is calculating 30-day back trajectories, by NOx rich airmasses. When these air masses have not been (or only in low probably) transported in the boundary layer in the last 30 days, we assume the relatively large NOx concentrations originated from lightning.

*Line 503: Can you briefly explain how "the probability of boundary layer influences" is obtained?*

**Response:** Clarified. The probability of boundary layer influences is provided by the ORNL DAAC (Ray et al., 2021) according to the percentage of time these air masses are located within the boundary layer for 30-day back trajectories. We have added the description about the dataset (line 234–235 page on 9).

We also use average probability of boundary layer influence in the dataset to identify air masses influenced by lightning (Sect. 3.4).

*Lines 523: The results for CAMS and MERRA-2 could be added to the Supplement.*

**Response:** Agreed and added. We have added the result of CAMS reanalysis to the supplement (Fig. S5).

Figure 14: How was this plot generated? Some of the colored points were previously categorized as e.g. urban influence and not as mixed pollution. Please clarify this. What are the gray data points? Maybe instead of the absolute values, the relative contribution to the overall O3 could be shown.

**Response:** Clarified. Figure 13 (previously Figure 14) describes ozone abundance associated with the biomass burning for all air masses classified as mainly "biomass burning", "mixed pollution" and "urban", which are the "polluted" categories. Gray circles are indicated for the other categories (stratosphere, marine and well-mixed & aged). This aspect is stated in the caption of the Figure 13.

We indicate in the text the relative contribution of $O_3$ to biomass burning, but not in the figure to avoid miss-leading indications of large relative biomass burning influence for air mass with very low ozone contributions.

*Figure 15: I recommend moving this Figure to the Supplement.*

**Response:** Agreed and done. We have moved this figure to the supplement (Fig. S6).

Line 637: This hypothesis could be tested by calculating ozone loss terms along the ATom2 flight track.

**Response:** Clarified. This comment is difficult to address since line 637 is the end of caption of the original Figure 16, but not a hypothesis. The reviewer might refer to the previous paragraph stating that "This is likely link to active sinks of tropospheric ozone during the transport from the continent due to the high relative humidity.". If this is the case, we find that ozone losses between the ocean and the continent (in the East-West direction) may not likely be estimated along the flight track (in the North-South direction) and therefore this analysis may not be done with the current dataset.

Lines 657 f.: This sounds like the reanalysis products only show overestimations for urban air masses? I understood earlier that the downward motions in the Hadley cell are overestimated in the reanalysis products leading to an overestimation in ozone. Could you please clarify this?

**Response:** Agreed and added. We agree that the overestimation of downward transport of ozone in the Hadley cell is also an interesting remark to add in the conclusions, and we did so (line 699–703 on page 37).

The two reanalyses show a stratospheric intrusion in the UTLS around 25° N, co-located with a descending branch of the Hadley cell, whereas IASI+GOME2 and ATom-3 do not show such a feature (the right panels of Fig. 7). In this region, we remark high potential vorticity and strong subsidence at altitudes between 8 and 15 km (Fig. S7a and c). However, relative humidity is high (Fig. S7e), which is not typical feature of stratospheric air. It suggests that the magnitude of downward transport of ozone from the stratosphere is likely overestimated by the reanalyses.

**Author Response to CC1 (Dr. Owen Cooper)**

*General comments:*

*This analysis evaluates three chemical reanalysis products above the tropical North and South Atlantic Oceans, with a focus on February 2017. Some discussion needs to be provided to state why the analysis only focuses on February 2017, when plenty of in situ observations are available for other seasons and years. For example, why not provide additional analysis of in situ observations for the months of September-October-November? This is the time of year when the well-known ozone maximum occurs above the South Atlantic (Thompson et al., 2021).*

**Response:** Agreed and added. Thank you for suggestion. Following the recommendation of the reviewers, we have added a whole new analysis for the period of October 2017 (corresponding to the ATom-3 campaign). This is large revision of the manuscript, increasing by a factor of 2 the volume and period of the year analyzed in the paper, covering indeed the spring season in the Southern Hemisphere as requested. Please, see page 1 of this document for more details.

*A paper recently published in the TOAR-II Community Special Issue (Gaudel et al., 2024) provides an extensive evaluation of several satellite products across the tropics. This paper takes advantage of 25 years of in situ observations, including several ATOM flights, 8 ozonesonde sites, and thousands of IAGOS aircraft profiles above five regions. These same data sets can be applied to your study. In particular the NASA SHADOZ ozonesonde station on Ascension Island reveals the seasonal ozone variability in the center of the South Atlantic ozone maximum (Thompson et al., 2021). This station also has several years (2016-2019) of surface ozone observations. The SHADOZ data archive is here: https://tropo.gsfc.nasa.gov/shadoz/Archive.html*

**Response:** Clarified. Due to a very limited coincident data within the campaigns (particularly with ATom-2), we have decided not to include SHADOZ observations for our study. Due to the significant number of analyses and figures already focusing on February and October 2017, we have avoided extending the length of the paper by including other periods from other years. Please see more details in the page 1 of this document.

*Finally, ozone production and transport above the tropical Atlantic has been studied for decades (Fishman et al., 1986, 1991; Thompson et al., 1996, 2000; Moxim and Levy, 2000), and it is important for the authors to describe their findings in relation to previous work, and to clearly state what is new about their findings.*

**Response:** Agreed and added. Thank you for this suggestion that we included in the revised manuscript. We have added the clear description of previous works according to the suggestion by Dr. Anne Thompson (CC4)**.**

*Specific Comments:*

*Lines 640-641*

*I don't understand the caption to Figure 16 as the sentence seems to be missing a verb.*

**Response:** Clarified. We have subtracted Fig. 16 for sake of briefness.

*Fishman et al. 1983 appears in the list of references, but I don't see this paper cited in the main text*

**Response:** Agree and suppressed from the reference list.

*This paper presents a study characterizing the vertical and horizontal distribution of tropospheric ozone over the South and Tropical Atlantic during February 2017 using a combined UV-IR product from IASI+GOME2, in situ airborne measurements from the Atmospheric Tomography Mission (ATom) and 3 reanalysis models.*

*The topic is important, but there are a few items that should be addressed as described below.*

*1) The observational data used within this study is very limited and should be expanded to include other vertical profiles available for your period of interest. Thompson et al (2021), which should be cited in this paper (and more recently Thompson et al (2025), presents a great reference for models and satellite products in the tropics, including the Atlantic, based on multiple SHADOZ ozonesonde stations within your region of interest. These stations include Ascension Island, UK, and Natal, Brazil, where vertical ozone profiles from 1998-2023 are available online here: https://tropo.gsfc.nasa.gov/shadoz/Archive.html. Also available during February 2017 is in situ ozone monitoring data from the Ascension Island station: https://tropo.gsfc.nasa.gov/shadoz/Ascension.html. It is important to provide insight from ground-based measurements for your tropospheric ozone discussion outside the limited aircraft flights and one satellite product.*

**Response:** Done and clarified. Thank you for your suggestions. We have revised Sect. 1 and added references including Thompson et al (2021) according to the comments by Dr. Anne Thompson (CC4).

We have decided not to include SHADOZ ozonesonde profiles for our study due to limited coincidence with the period of the field campaigns (particularly ATom-2). Please see more details in page 1 of this document.

Moreover, we mainly on ozone distribution at higher altitudes within the troposphere. Therefore, and for sake of briefness, we have not included surface ozone observations at a single location (Ascension Is. Station).

*2) Recent studies using ground-based and satellite observations discussing current tropical tropospheric ozone distributions should be addressed and cited: Thompson et al (2021; 2025) and Gaudel et al. (2024).*

**Response:** Cited.

3) Based on the knowledge that UV/IR satellite products are limited with their sensitivity in the lower troposphere (using 0-3km layer here) and over water (versus land), have you done comparisons with similar tropospheric ozone satellite products? A demonstration of what the IASI-GOME2 vertical profiles, and other similar satellite products, look like over a tropical Atlantic site like Ascension Island for your period of interest would show the full extent of the vertical ozone distribution and any limitations in the presented satellite and model reanalysis. Pennington et al (2024) describes a joint AIRS/OMI ozone product similar to the IASI-GOME2 that could be used and Keppens et al (2025) provides a list of current tropospheric ozone observing satellite products that could be compared here.

**Response:** Clarified. Including additional satellite products would clearly enlarge the length of the paper, in terms of figures and comments to discuss the differences. This satellite product comparison issue is a key point for papers such as Gaudel et al. (2024 ACP) or Keppens et al. (2025, ACPD) which already include IASI+GOME2 and other datasets. However, our papers is focusing on the ozone distribution and its origin, thus using already multiple datasets of different nature such as in situ detailed measurements, satellite data and chemical reanalysis. Thus, for sake of briefness, we have not included other ozone satellite products with IASI+GOME2. Moreover, AIRS+OMI product is available only 2021–2025 on the GES DISC thus not coincident with ATom-2 and ATom-3.

**Author Response to CC3 (Dr. Kenneth Pickering)**

*Major comments:*

*I do not think that the authors should have included MERRA-2 in this analysis. There is no detailed tropospheric ozone photochemistry included in the model -- only a simple month-dependent O₃ parameterization (Stanjer et al., 2008, JGR) derived from a 2-D chemistry model that was originally developed for the stratosphere. Monthly mean O₃ production and loss terms from the 2-D model are utilized in the GEOS-5 model, and these terms are not really useful below the upper troposphere. As a result, O₃ in the lower and middle troposphere is not well represented in the assimilation, especially with regard to longitudinal variations in O3. The MERRA-2 developers do not recommend use of the O₃ values below 500 hPa.*

**Response:** Agreed and withdrawn. Thank you for your suggestion. We have excluded MERRA-2 from this intercomparison study.

*Section 2.2 on the Atmospheric Chemistry Reanalyses: The descriptions of the three reanalyses do not say how lightning flashes are predicted in the models or say how lightning NOx (LNOx) emissions are treated (NOx production per flash and how those emissions are distributed vertically in the model). Therefore, the reader does not know how realistic the distribution of LNOx emissions is represented in the model. This horizontal and vertical distribution will have a large impact on the source attribution results for O₃. LNOx is not included at all in MERRA-2, significantly affecting the longitudinal distribution of O₃ especially in the upper troposphere. Assimilation of OMI and MLS O₃ will partially correct for this deficiency, but not totally. Lack of LNOx leads to a small low bias in UT O₃ in MERRA-2.*

**Response:** Agreed and added. Thank you for your suggestion. The suggestion has notably improved Section 2.2 of our paper. We have added the descriptions about the lightning emissions (lines 168–172, lines 186–195 on page 7 and Table 1).

Lightning $NO_X$ ($LNO_X$) sources were simulated by using the convection scheme of MIROC-AGCM (Miyazaki et al., 2017). The global distribution of the flash rate was parameterized for convective clouds based on the relationship between lighting activity and cloud top height (Price and Rind, 1992). The vertical profile of the $LNO_X$ sources were determined on the basis of the C-shape profile, which peaks at the surface and in the upper troposphere, given by Pickering et al. (1998).

$LNO_X$ emissions are simulated by the modules for atmospheric composition in the IFS, named Composition-IFS (C-IFS) (Flemming et al., 2015). The C-IFS has two options to simulate the flash-rate densities using the following input parameters: (i) convective cloud height (Price and Rind, 1992) or (ii) convective precipitation (Meijer et al., 2001). Flemming et al. (2015) showed the comparison of the annual flash rate density from the IFS input data using the parameterisation by Price and Rind (1992), Meijer et al. (2001) and observations. The smaller land–sea differences of Meijer et al. (2001) agreed better with the observations. The observed maximum over central Africa was well reproduced by both parameterisations, while an exaggerated maximum was remarked over tropical South America. In the IFS, $LNO_X$ emissions uses the parameterisation of Meijer et al. (2001) based on convective precipitation. The vertical profile of the $LNO_X$ sources were determined on the basis of the backward C-shape profile, which locates most emission in the middle of the troposphere, given by Ott et al. (2010).

Specific comments:

*Abstract line 22-23: most of the lift occurs over the continents, not the ocean*

**Response:** Agreed and corrected (line 22 page 1).

These datasets show air masses enriched in ozone precursors from biomass burning sources over Western and Central Africa lifted into the middle and upper troposphere by strong upward motions; and stratospheric intrusions in the descending branches of the Hadley cells over the Southern Atlantic.

Figure 1: shows 2017 maps of the World-Wide Lightning Detection Network (WWLLN) Global Lightning Climatology (WGLC) from Kaplan and Lau (2021, Earth Sys. Sci. Data). Much greater maximum amounts of lightning are shown over Central and South America than over Africa. The satellite (OTD/LIS) climatology (Cecil et al., 2014, Atmos. Res.) is much different, showing the strongest maximum over Central Africa. WWLLN detection efficiency over Africa is very low. I'm wondering if enough attention has been given to this when the WGLC was created. Uncertainty in use of this climatology should be mentioned in the paper.

**Response:** Agreed and added. We have added the description about the difference between the WGLC and the LIS/OTD discussed in Kaplan and Lau (2021) (lines 253–259 on page 9–10).

Kaplan and Lau (2021a) compared climatological mean annual lightning density between the WGLC and the LIS/OTD, even though the periods of the record are not overlapping and the lightning phenomenon observed, i.e., strokes in the WGLC and flashes in the LIS/OTD, is different. The LIS/OTD captured more lightning than the WGLC, particularly over land. The area of the greatest difference was in the eastern Congo Basin, this was also a hotspot for lightning in the WGLC. Other regions where the WGLC had lower lightning than the LIS/OTD were in the Western High Plateau of Cameroon and northwestern South America. In the Northern Hemisphere and over the oceans, the differences were smaller.

*Figures 4, 6, and others: MERRA-2 $O_3$ plots for 0-3 km should not be shown here – $O_3$ is obviously wrong in the lower troposphere*

*Figures 5 and 7: MERRA-2 $O_3$ plots for a specific day should not be trusted and therefore not shown. The chemistry in the model is based on monthly mean O3 production and loss rates.*

**Response:** Agreed and withdrawn. We have excluded MERRA-2 from this study.

**Author Response to CC4 (Dr. Anne Thompson)**

*The paper as a whole follows a reasonable outline and is well-organized but it begins with serious omissions that must be remedied. It is easy to forget the pioneering satellite work or major experiments that paved the way for ATom or HIPPO (Wofsy: https://royalsocietypublishing.org/doi/10.1098/rsta.2010.0313 ), a similar experiment that preceded the ATom Atlantic and Pacific transects. The SAFARI-92/TRACE-A sampling over the Atlantic and similar DC-8 campaigns over the Pacific provided baseline data and the insights into tropical chemistry and dynamics that remain relevant.*

**Lines 40 ff. Context with prior published work. The south and tropical Atlantic is NOT one of the regions "with the sparest coverage of in-situ observations!"** *On the contrary, south Atlantic ozone structure is well-characterized and has been studied in detail since the early 1990s. Before that a classic paper by Jack Fishman on satellite data from TOMS (https://agupubs.onlinelibrary.wiley.com/doi/abs/10.1029/JD092iD06p06627) called attention to the high tropospheric ozone over the south Atlantic. That was the beginning of "Tropical Atlantic" studies on ozone. A number of satellite-based products have focused on the tropical south Atlantic, e.g., Fishman et al., JGR, 1996; Thompson et al., GRL, 2000; Hudson et al., JGR, 1995; Leventidou et al., ACP, 2017.*

*Just as important, pioneering international aircraft campaigns led by the US, France and Germany probed causes of the "tropical south Atlantic ozone maximum" with flights from Brazil, western and southern Africa and from Brazil. The early aircraft campaigns were:*

- *TROPOZ II – led by A. Marenco from Toulouse!! For example: https://www.sciencedirect.com/science/article/abs/pii/S1352231099005087 by Gouget. Also - https://www.semanticscholar.org/paper/Study-of-ozone-formation-and-transatlantic-from-the-Jonqui%C3%A8res-Marenco/fcdb685886daa73cd211fb1e3e740d5138d2c02a*

- *Early IGAC "Projects" on biomass burning and south tropical Atlantic (BIBEX, STARE). In September-October 1992 a massive umbrella experiment "STARE" was conducted during the southern African burning season: IGAC/STARE/SAFARI-92/TRACE A (International Global Atmospheric Chemistry/South Tropical Atlantic Regional Experiment/Southern African Fire Atmospheric Research Initiative/Transport and Atmospheric Chemistry near the Equator-Atlantic) with multiple aircraft, including NASA's DC-8 with comprehensive lidar ozone measurements, in-situ profiling of ozone precursors like NO/NO2, CO, hydrocarbons; SAFARI-92 operated a DC-3 over southern Africa. The aircraft measurements were augmented by balloon ozonesonde profiles from Brazil, Ascension Island, Brazzaville, Congo; Irene, South Africa. There are ~65 papers from the 1992 campaigns in Journal of Geophysical Research, published 1 Oct. 1996!*

*An outgrowth of the SAFARI-92/TRACE-A balloon data was the initiation of the SHADOZ (Southern Hemisphere ADditional Ozonesondes) network of ozone sounding stations that has operated from 1998 to the present day (Thompson et al., 2003; Thompson et al., 2017). SHADOZ provides ozone profiles 2-4 times/month from Ascension Island (central south tropical Atlantic, 8S, 15W), Natal, Brazil (6S, 35W) and Paramaribo (5N, 55W). The ozone profiles from these 3 stations since 1998 number more than 2000!*

**The earlier work must be cited. One way to revise the paragraph starting on Line 40 …**

*The South and Tropical Atlantic has been a region of intense interest in the ozone community since a regional maximum in tropospheric ozone derived from satellite measurements was identified by Fishman and Larson (1987). This discovery was the motivation for a large-scale ground and aircraft study in the southern biomass burning season in September and October 1992: IGAC/STARE/SAFARI-92/TRACE A (International Global Atmospheric*

*Chemistry/South Tropical Atlantic Regional Experiment/Southern African Fire Atmospheric Research Initiative/ Transport and Atmospheric Chemistry near the Equator-Atlantic). SAFARI-92/TRACE A confirmed the regional ozone feature with aircraft profiling and lidar plus ozonesondes deployed over Brazil, Ascension Island and 3 sites in subSaharan Africa. In addition, analyses of the comprehensive SAFARI-92/TRACE-A data confirmed links of the Atlantic maximum to fire activity over Africa (Fishman et al., 1996; Thompson et al., 1996) and to ozone formed from a combination of fires, deep convection and lightning activity over South America (Pickering et al., 1996). Based on the ozonesonde profiles, it was estimated that the relative contributions to the Atlantic ozone were approximately 2/3 from African sources and 1/3 from South America (Thompson et al., 1996). However, dynamical influences were required for the ozone feature to form. Krishnamurti et al. (1996) demonstrated that recirculation within the south Atlantic gyre allowed the ozone to accumulate so that the highest ozone amounts were over the ocean rather than the continents.*

*Shipboard ozone sampling over the tropical Atlantic provided additional insights into south tropical Atlantic ozone (Weller et al., 1996; Thompson et al., 2000). The ozone maximum occurred at all seasons, not only during the peak of southern hemisphere burning but also when African fire activity was at its greatest north of the ITCZ. This so-called "Atlantic ozone paradox" was associated with upper tropospheric-stratospheric subsidence and lightning in addition to fires (Thompson et al., 2000). These contributions were evaluated in an early model study (Moxim and Levy, 2000). The SAFARI-92/TRACE-A experiments were instrumental in assembling the SHADOZ (Southern Hemisphere Additional Ozonesondes; https://tropogsfc.nasa.gov/shadoz) network of stations that has operated from 1998 to the present day (Thompson et al. 2017). With coordinated launches of ozonesondes from more than 10 stations across the tropics, the Atlantic maximum is a strong feature with the south tropical Atlantic always exhibiting more tropospheric column ozone (5-15 Dobson Units,1 DU = 2.69x10$^{16}$ cm$^{-2}$). When looking at tropospheric ozone structure across the entire tropical band, the Atlantic ozone feature leads to a zonal wave-one pattern (Thompson et al., 2003).*

*More recently, the role of biomass burning (van der Werf et al., 2017), biogenic (Sindelarova et al., 2022) and lightning (Schumann and Huntreiser, 2007) contributions to Atlantic, African and South America ozone has been investigated. In-Service Aircraft….*

**Response:** Agreed and added. We appreciate a lot the suggestion. We also apologize for the lack of literature review in the previous version of the manuscript. We have added the suggested descriptions. This much more complete information has notably improved Sect. 1.

*Comment on Figs 9 and 10. Comparisons of satellite and model ozone with April 2017 SHADOZ sondes should be carried out for comparison to the ATOM comparisons.*

**Response:** Clarified. We have decided not to include SHADOZ observations for our study due to lack of sufficient co-located data with the ATom field campaigns (specially for ATom-2) and sake of briefness. Please see more details in page 1 of this document.